# Deep learning predicts cardiovascular disease risks from lung cancer screening low dose computed tomography

Hanqing Chao [1], Hongming Shan [1], Fatemeh Homayounieh [2], Ramandeep Singh [2], Ruhani Doda Khera[2], Hengtao Guo[1], Timothy Su[3], Ge Wang [1✉], Mannudeep K. Kalra [2✉] & Pingkun Yan [1✉]

Cancer patients have a higher risk of cardiovascular disease (CVD) mortality than the general population. Low dose computed tomography (LDCT) for lung cancer screening offers an opportunity for simultaneous CVD risk estimation in at-risk patients. Our deep learning CVD risk prediction model, trained with 30,286 LDCTs from the National Lung Cancer Screening Trial, achieves an area under the curve (AUC) of 0.871 on a separate test set of 2,085 subjects and identifies patients with high CVD mortality risks (AUC of 0.768). We validate our model against ECG-gated cardiac CT based markers, including coronary artery calcification (CAC) score, CAD-RADS score, and MESA 10-year risk score from an independent dataset of 335 subjects. Our work shows that, in high-risk patients, deep learning can convert LDCT for lung cancer screening into a dual-screening quantitative tool for CVD risk estimation.

---

[1] Department of Biomedical Engineering, Biomedical Imaging Center, Rensselaer Polytechnic Institute, Troy, NY, USA. [2] Department of Radiology, Massachusetts General Hospital, Harvard Medical School, Boston, MA, USA. [3] Niskayuna High School, Niskayuna, NY, USA. ✉email: wangg6@rpi.edu; mkalra@mgh.harvard.edu; yanp2@rpi.edu

Cardiovascular disease (CVD) affects nearly half of American adults and causes more than 30% of fatality[1]. The prediction of CVD risk is fundamental to the clinical practice in managing patient health[2]. Recent studies have shown that the patients diagnosed with cancer have a ten-fold greater risk of CVD mortality than the general population[3]. For lung cancer screening, low dose computed tomography (LDCT) has been proven effective through clinical trials[4,5]. In the National Lung Screening Trial (NLST), 356 participants who underwent LDCT died of lung cancer during the 6-year follow-up period. However, more patients, 486 others, died of CVD. The NELSON trial shows similar overall mortality rates between the study groups even though the lung cancer mortality decreased in the LDCT screening group[6]. Therefore, screening significant comorbidities like CVD in high-risk subjects undergoing LDCT for lung cancer screening is critical to lower the overall mortality. Nevertheless, when the cancer risk population receives cancer screening, their potential CVD risk may be overlooked. A recent study reported that only one-third of patients with significant coronary artery calcification (CAC) on LDCT had established coronary artery disease diagnosis, whereas just under one-quarter of the patients had a change in his/her cardiovascular management following LDCT with referral to cardiologists[7].

Since the Medicare coverage for LDCT lung cancer screening started in 2015 in the United States, the use of LDCT in eligible high-risk subjects has increased dramatically, with 7–10 million scans per year[8]. Most subjects eligible for lung cancer LDCT screening often have an intermediate to high risk for CVD[9]. It is of great importance for this high-risk population to have an additional CVD screening. The clinical standard requires a dedicated cardiac CT scan to estimate the CVD risk, which induces a significant cost and high radiation. In contrast, chest LDCT has been shown to contain information especially CAC strongly associated with the CVD risk[10], but there is no consensus of using LDCT images for CVD risk assessment due to the low signal-to-noise ratio and strong image artifacts. This suggests opportunities for advanced systems to tackle the limitations and predict CVD risks from LDCT images.

In the past decade, machine learning, especially deep learning, has demonstrated an exciting potential to detect abnormalities from subtle features of CT images[11]. Several machine learning methods were proposed to estimate CVD factors automatically from CT images. A majority of those methods predict clinically relevant image features including CAC scoring[12–17], non-calcified atherosclerotic plaque localization[18–22], and stenosis [23–27] from cardiac CT. For LDCT images, subject to motion artifacts and low signal-to-noise ratio in contrast to cardiac CT images, only until recently were deep learning algorithms applied to quantify CAC scoring from LDCT images as a surrogate of the CVD risk[28–31]. Even fewer existing methods directly estimate the CVD risk from LDCT images. van Velzen et al.[32] proposed a two-stage method to predict cardiovascular mortality: first extracting image features using a convolutional autoencoder, and then making a prediction using a separate classifier such as a neural network, random forest, or support vector machine. However, such a two-stage method may not be able to extract distinctive features associated with CVD, since the first stage has little knowledge about the final objective. Our previous work showed the feasibility of predicting all-cause mortality risk from patient's LDCT images from 180 subjects[33]. The developed method, KAMP-Net, first selects a representative 2D key slice from the whole LDCT volume, and then applies an end-to-end CNN to predict all-cause mortality risk (area under the curve (AUC) of 0.76).

To tackle the limitations of the prior studies, we propose an end-to-end deep neural network to (a) screen patients for CVDs and (b) quantify CVD mortality risk scores directly from chest LDCT examinations. Specifically, our approach focuses on the cardiac region in a chest LDCT scan and makes predictions based on the automatically learned comprehensive features of CVDs and mortality risks. The prediction is calibrated against the incidence of CVD abnormalities during the follow-up period of a clinical trial, subjective assessment of radiologists in reader studies, and the CVD risk scores calculated from electrocardiogram (ECG)-gated cardiac CT including the CAC score[34], CAD-RADS score[35], and MESA 10-year risk score[36].

Figure 1 shows an overview of our study. Two datasets with a total of 10,730 subjects were included in our study (Fig. 1a). The public National Lung Screening Trial (NLST) dataset was used for model development and validation. It includes lung cancer screening LDCT exams of 10,395 subjects with abnormality records from the exam reports and causes of death for deceased subjects. An independent dataset collected at Massachusetts General Hospital (MGH) was used for independent validation. Besides images and clinical reports of LDCT exams, the MGH dataset also collected ECG-gated cardiac CT of the same group of subjects, which enables us to calculate the clinically used CVD risk scores for validation. Our approach consists of two key components. First, a CNN heart detector was trained with 263 LDCTs from the NLST dataset to isolate the heart region (Fig. 1b). Second, we propose a three-dimensional (3D) CNN model, Tri2D-Net, consisting of a CVD feature extractor and a CVD screening classifier. We trained Tri2D-Net using CVD screening results as targeted labels (Fig. 1c). After training, the predicted probability of being CVD positive is used as a quantified CVD mortality risk score, which was validated by the CVD mortality labels on the NLST dataset. To further evaluate the generalization capability of our model, we calibrated the learned high-dimensional CVD features with three popular gold-standard CVD risk scores, including CAC score[34], CAD-RADS score[35], and MESA 10-year risk score[36].

## Results

**Datasets**. In total, the NLST and MGH datasets included 6276 males and 4454 females aging from 37 to 89, forming a population of 10,730 subjects. Details of the datasets are provided as follows.

NLST enrolled in a total of 26,722 subjects in the LDCT screening arm. Each subject underwent one to three screening LDCT exams, each of which contains multiple CT volumes generated with different reconstruction kernels. We received 47,221 CT exams of 16,264 subjects from NCI, which has reached the maximum number of subjects allowed for a public study. We labeled each subject in the dataset as either CVD-positive or CVD-negative to conduct CVD screening on this dataset. A subject was considered CVD-positive if any cardiovascular abnormality was reported in the subject's CT screening exams or the subject died of CVD. A CVD-negative subject has no CVD-related medical history and no reported cardiovascular abnormality in any CT scans during the trial and did not die of circulatory system diseases. Among the available dataset, there are 17,392 exams used in our study from 7433 CVD-negative subjects and 4470 exams from 2962 CVD-positive subjects. The subjects were randomly split into three subsets for training (70% with 7268 subjects), validation (10% with 1042 subjects), and testing (20% with 2085 subjects). Supplementary Fig. 2 shows the detailed inclusion/exclusion criteria and the resultant distribution of LDCT exams in the three subsets. Tables 1 and 2 list the characteristics of the dataset (for more details, see Methods, NLST dataset). To quantify CVD mortality risk, we identified

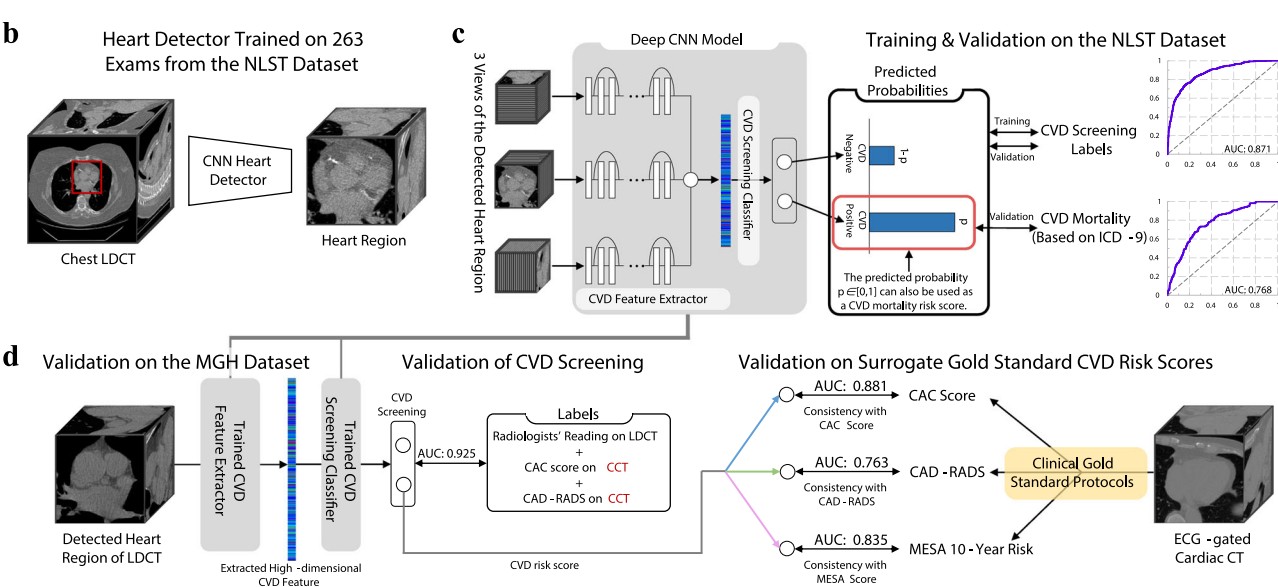

**Fig. 1 Overview of the proposed deep learning workflow. a** Public NLST and independent MGH datasets. The MGH dataset contains both LDCT and ECG-gated cardiac CT (CCT) images of 335 subjects. **b** To extract an image volume of the heart, a deep CNN heart detector was developed and trained. **c** A 3D CNN, Tri2D-Net, was designed for simultaneously screening CVD and predicting mortality risk. After training, the output probability of CVD positive also quantifies CVD mortality risk. **d** The proposed model was further validated on the MGH dataset.

**Table 1 Demographics of the two independent datasets used in this study.**

| Dataset | NLST LDCT | | | MGH | |
| | CVD screening positive/negative | CVD-related deaths/survival | Overall | LDCT | Cardiac CT |
|---|---|---|---|---|---|
| # Patients | 2962/7433 | 600/9795 | 10,395 | 335 | 235 |
| Men | 2048/4067 | 442/5673 | 6115 | 161 | 125 |
| Women | 914/3366 | 158/4122 | 4280 | 174 | 110 |
| Age (years) | 62.9 ± 5.3/60.8 ± 4.8 | 64.0 ± 5.5/61.3 ± 5.0 | 61.4 ± 5.0 | 63.6 ± 8.0 | 64.9 ± 7.8 |
| Weight (kg) | 85.9 ± 19.3/79.4 ± 17.1 | 86.5 ± 21/81.0 ± 17.7 | 81.3 ± 18.0 | 82.9 ± 19.0 | 83.6 ± 18.5 |

**Table 2 CT Scan characteristics of the two independent datasets used in our study.**

| Dataset | NLST LDCT | MGH | |
| | | LDCT | Cardiac CT |
|---|---|---|---|
| Tube voltage (kVp) | 120/130/140 | 120/100 | 120 |
| Milliampere-seconds (mAs) | 58.06 ± 20.59 | 33.80 ± 11.53 | 37.44 ± 10.06 |
| Slice thickness (mm) | 2.3 ± 0.4 | 1.0/1.25 | 3.0 |
| Slice overlap (mm) | 2.0 ± 0.4 | 0.8/1.0 | 1.5 |
| In-plane resolution (mm) | 0.66 ± 0.07 | 0.83 ± 0.09 | 0.34 ± 0.04 |
| Helical pitch | 1.4 ± 0.2 | 1.1 ± 0.2 | Axial mode |

CVD-related mortality based on the ICD-10 codes provided in the dataset. The selected ICD-10 codes are shown in Supplementary Table 1. Note that all subjects with CVD-related causes of death were considered CVD-positive.

Furthermore, through an institutional review board (IRB) approved retrospective study, we acquired an independent and fully deidentified dataset from MGH in 2020. This MGH dataset contains 335 patients (161 men, 174 women, mean age 63.6 ± 8.0 years), who underwent LDCT for lung cancer screening. In this dataset, 100 patients had no observed CVD abnormalities in their LDCT. The remaining 235 subjects underwent ECG-gated cardiac CT for CVD risk assessment due to atypical chest pain, equivocal stress test, and chest pain with low to intermediate risk for heart diseases. Three CVD risk scores were calculated for the 235 subjects from their cardiac CT images, including CAC score[34], coronary stenosis (quantified as CAD-RADS)[35] and MESA 10-year risk[36]. Tables 1 and 2 list the characteristics of the dataset (see Methods, MGH dataset). The MGH dataset was used to evaluate the clinical significance of the NLST-trained model for feature extraction without re-training or fine-tuning. A subject would be labeled as CVD-positive if the subject underwent an ECG-gated cardiac CT screening and received either a CAC score > 10 or a CAD-RADS score > 1. Correspondingly, a CVD-negative subject had either all the LDCT screening exams being

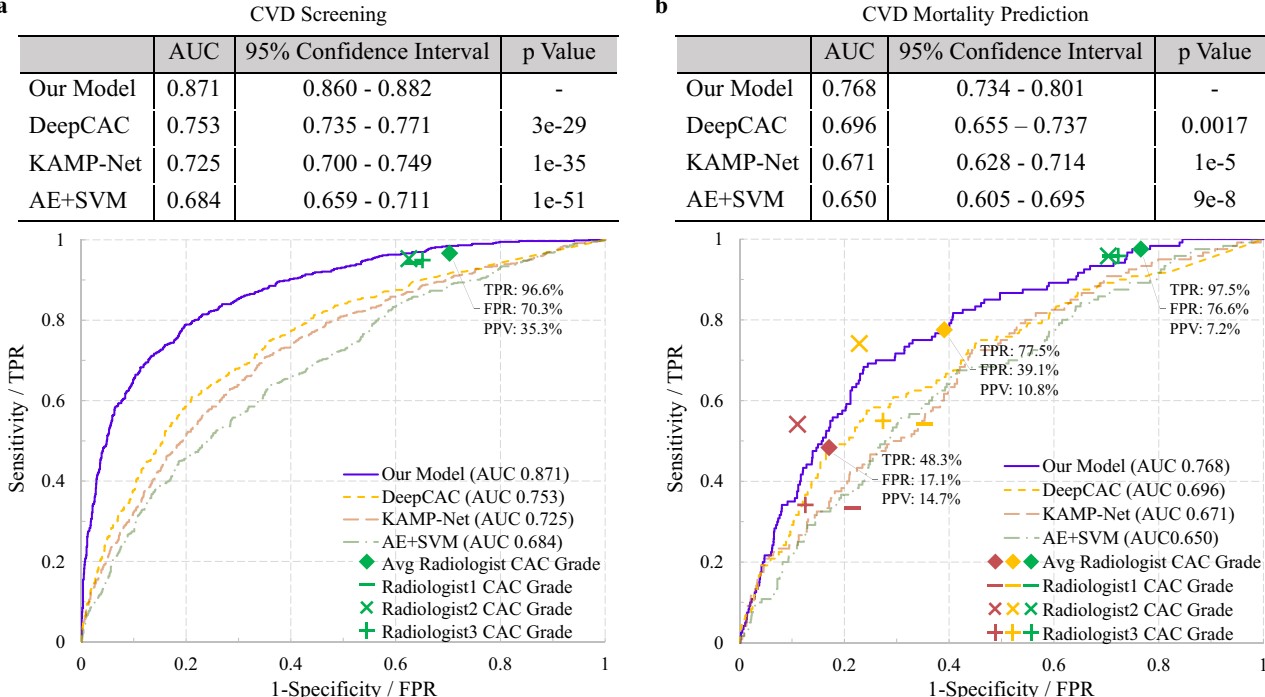

**Fig. 2 Experimental results on the NLST dataset.** Comparison of our model with radiologists and other reported methods on (**a**) CVD detection and (**b**) CVD-caused mortality prediction. For the reader study, green, yellow and red symbols indicate the performance of CVD risk estimation using CAC Grades 1+, 2+, and 3, respectively. *p* values were computed using a one-tail z-test.

negative or the calculated CAC scores ≤10 and CAD-RADS ≤1. Based on the above criteria, 181 subjects were CVD positive, and 154 patients were CVD negative. To quantify CVD mortality risk, since this MGH dataset was collected from a recent patient with exams performed between 2015 and 2020, the mortality records are not available. We instead calibrate our model against the three gold-standard risk scores as surrogate evaluators.

**Retrospective findings on NLST.** Two experiments were conducted on the NLST dataset for the evaluation of CVD screening and CVD mortality quantification, respectively, where the proposed deep learning model was compared with other deep learning models and against CAC grades read by radiologists. Three radiologists from MGH (M.K.K., R.S., and R.D.K.) with 2–15 years of clinical experience averaged at 7 years, independently graded all the 2085 CT volumes to obtain the CAC grades. Four CAC categories were used, including no calcification (level 0—normal), calcification over < 1/3 of the length of coronary arteries (level 1—minimal), calcification over 1/3 to 2/3 of the coronary arterial lengths (level 2 —moderate) and calcification > 2/3 of the arterial length (level 3— heavy). The average CAC grade is calculated by averaging the CAC grades read by the three radiologists.

We first evaluated the proposed model for identifying patients with CVDs from the lung cancer screening population. Figure 2a shows the receiver operating characteristic curves (ROCs) of multiple methods. Our deep learning model achieved an area under the curve (AUC) of 0.871 (95% confidence interval, 0.860-0.882)(see "Methods", Statistical analysis). With a positive predictive value (PPV) of 50.00%, the model achieved a sensitivity of 87.69%, which suggests that our model can identify 87.69% of the CVD-positive subjects using only a chest LDCT scan, when allowing half of the positive predictions as false. For the radiologist performance, all patients with ≥ minimal CAC

(CAC Grade 1+) are considered as abnormal. It can be seen in Fig. 2a that CAC Grade 1+ yielded a sensitivity of 96.6% and a PPV of 35.3%. With a similar sensitivity of 96.6%, our model achieved a slightly but not significantly higher PPV of 38.4% (*p* = 0.3847). In addition, we compared our model with the three recently reported works, KAMP-Net[33], Auto-encoder (AE +SVM)[32], and a deep learning based CAC socring model (DeepCAC)[17]. The table in Fig. 2a shows that our model significantly outperformed the other three methods (*p* < 0.0001). It indicates that our model can use LDCT to differentiate subjects with high CVD risk from those with low risk.

Further, we evaluated the performance of our model in quantifying CVD mortality risk. The results are shown in Fig. 2b, where CAC Grades 1+, 2+, and 3 denote the performance of mortality prediction using the extent of subjective CAC categories 1 and above, categories 2 and above, and 3 only, respectively. The trained deep learning model was directly applied to this testing set to predict the CVD-caused mortality without fine-tuning. Our deep learning model achieved an AUC value of 0.768 (95% confidence interval, 0.734–0.801), which significantly outperformed the competing methods (*p* < = 0.0017) as shown in Fig. 2b. With the same PPV of averaged CAC Grade 2+ (10.8%), our model achieved a sensitivity of 80.8%. Specifically, in the NLST test set, our model successfully identified 97 of the 120 deceased subjects as high risk, while the averaged CAC Grade 2+ labeled 35 of those 97 cases as low risk. Additionally, it can be seen from Fig. 2b that our model achieved similar performance to the average performance of human experts. It is worth mentioning that a significant difference exists between the three radiologists' annotations (*p* < 0.0001), even though radiologist 1 performed better than our model. For further comparison, Fig. 3 shows the Kaplan–Meier curves of different risk groups labeled by our model and the radiologists, respectively. For the radiologists, we used the average radiologist prediction of CAC Grade 2+ to

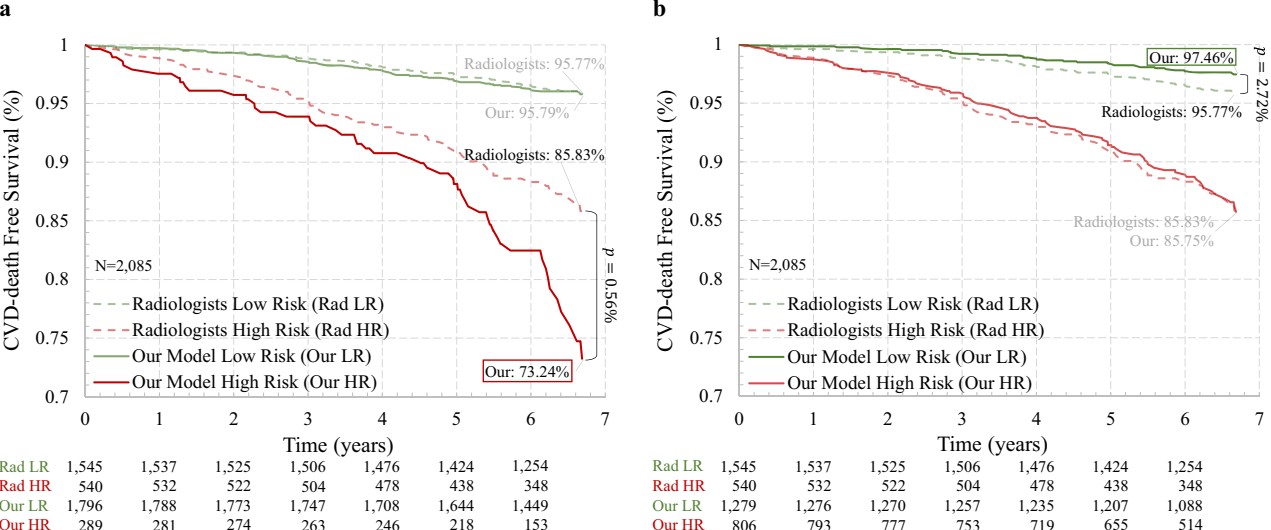

**Fig. 3 Kaplan–Meier curves on the NLST dataset.** Comparison of our model with the radiologists. Thresholds were selected on the CVD risk score calculated by our model to enforce the low (**a**) or high (**b**) risk group has a similar survival probability with that of the radiologists.

separate the subjects into low/high risk groups and drew the Kaplan–Meier curves for both groups. The final survival probabilities of low and high risk groups by radiologists are 95.79% and 85.83%, respectively. For fair and direct comparison, in Fig. 3a, we selected a threshold to divide the quantified CVD risk scores using our model so that the low risk group has a survival probability of 95.79%, similar to the radiologists. Under this circumstance, the model-predicted high-risk group showed a significantly lower ($p = 0.0059$) final survival probability of 73.24%. Similarly, in Fig. 3b, we selected a threshold so that the high-risk group has a survival probability of 85.75%, also similar to the radiologists. In this case, the model-predicted low-risk group achieved a significantly higher ($p = 0.0272$) final survival probability of 97.46%. Thus, our model can help reduce inter-and intra-observer variations in quantifying CAC. The model can also automatically categorize CVD risks so that radiologists can focus on other tasks such as lung nodule detection, measurement, stability assessment, classification (based on nodule attenuation), and other incidental findings in the chest and upper abdomen.

**Validation on the Independent MGH dataset**. To investigate the generalizability of our AI model, we directly applied the model trained on the NLST dataset to the MGH dataset. Four experiments were conducted including one experiment for the validation of CVD screening and three experiments evaluating the reliability of the deep learning model against gold-standard CVD risk factors (Fig. 1d).

In the experiment of CVD screening shown in Fig. 4a, our deep learning model achieved a significantly higher ($p < 0.0001$) AUC value of 0.924 (95% confidence interval, 0.909–0.940) than its performance on the NLST dataset (0.871), where the network was originally trained. Superior performance on this external MGH dataset may be due to the following two factors. First, MGH dataset acquired with contemporary scanners contains better quality images. Second, the MGH dataset combines the annotations of LDCT and ECG-gated cardiac CT as gold standard, which is more accurate than the annotation in NLST

To evaluate the generalization ability of the deep learning quantified CVD risk score, we directly applied the trained model to the MGH data and evaluated the consistency between the

model-predicted risk score from LDCT and the three clinically adopted risk scores calculated from ECG-gated cardiac CT. Our model was compared with two other previously reported studies on CVD risk prediction[32,33].

The predicted risk score from LDCT was first evaluated against the CAC score[34]. With a threshold of 400 for CAC scores, the MGH subjects were divided into two groups: 78 subjects with severe CAC and 157 subjects with none or minor CAC. Our model achieved an AUC value of 0.881 (95% confidence interval, 0.851–0.910) and significantly outperformed the other two methods ($p < 0.0001$, see Fig. 4b), despite the fact that our model has never been trained for CAC score estimation. These results suggest that our deep learning quantified CVD risk score is highly consistent with the CAC scores derived from ECG-gated cardiac CT in differentiating patients with severe and non-severe CAC.

The second experiment evaluates the capability of the deep learning model in classifying subjects into high and low-risk groups using LDCT by comparing against the coronary stenosis (CAD-RADS) scores[35] obtained by human experts on CCT. Subjects with CAD-RADS scores greater than or equal to 4 are labeled as with severe stenosis, i.e., positive samples (51 subjects). The other 184 subjects with smaller scores were labeled as negative. Our model reached an AUC value of 0.763 (95% confidence interval, 0.704–0.821, see Fig. 4c). Our model significantly outperformed the other two methods ($p \leq 0.0080$). Unlike calcification, coronary stenosis is much harder to detect through a chest LDCT screening, while it is a direct biomarker of CVD risk. The performance obtained using LDCT is thus highly encouraging. The superiority demonstrates that our model can quantify the subclinical imaging markers on LDCT, making it a promising tool for CVD assessment in lung cancer screening.

In the third experiment, patients were divided into high and low-risk groups according to MESA 10-year risk score[36], which is a clinical gold-standard risk stratification score for CVD integrating multiple factors including gender, age, race, smoking habit, family history, diabetes, lipid lowering and hypertension medication, CAC score extracted from CCT, and laboratory findings including cholesterol and blood pressure. Because some of the 235 subjects did not have all the needed exams, we are only able to calculate the MESA scores of 106 subjects. When median MESA 10-year risk score in our patients was used as a threshold

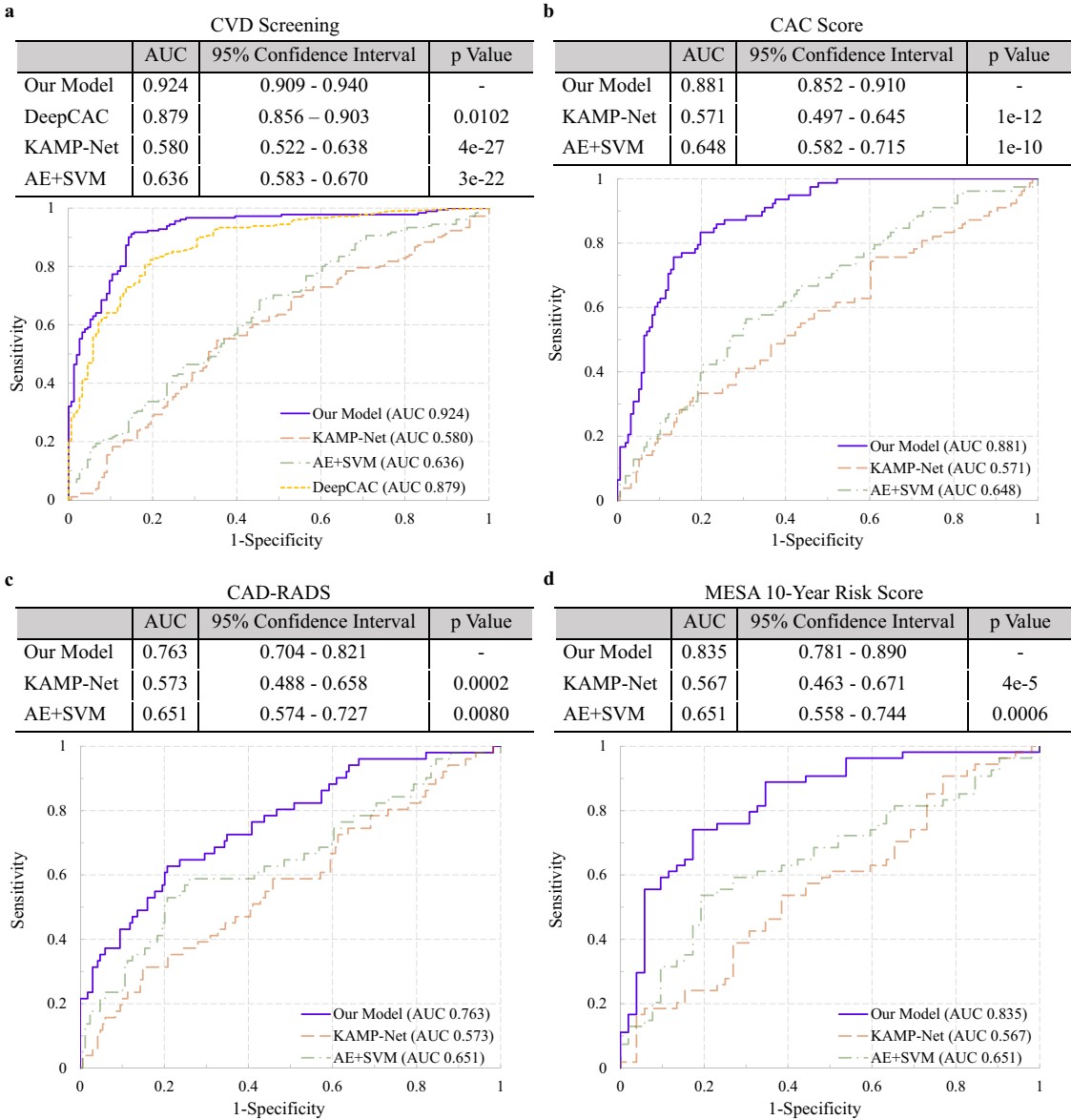

**Fig. 4 Results of the four experiments on the MGH dataset.** (**a**) Validation of CVD screening. **b**–**d** Comparison of our deep learning model with three clinically used risk scores, including (**b**) CAC score[34], (**c**) CAD-RADS[35], and (**d**) MESA 10-year risk score[36]. *p* values were computed using a one-tail z-test.

(14.2), 52 subjects with greater scores were labeled as high risk, while the other 54 subjects were labeled as low risk. Our model achieved an AUC value of 0.835 (95% confidence interval, 0.781–0.890), which significantly outperformed all the other methods (see Fig. 4d).

## Discussion

In summary, our deep learning model demonstrates the value of lung cancer screening LDCT for CVD risk estimation. Given the increasing utilization of LDCT-based lung cancer screening, shared risk factors, and high prevalence of CVD in these at-risk patients, the potential of obtaining a quantitative and reliable CVD risk score by analyzing the same scans may benefit a large patient population. The further comparative study on the deep learning model of LDCT images with human experts on CCT for risk group classification shows that the deep learning model can analyze LDCTs to achieve performance approximating the clinical reading with dedicated cardiac CTs. The comparable or

superior performance of our model from LDCT implies that additional dedicated ECG-gated coronary calcium scoring and other laboratory tests could be avoidable. Our deep learning model may thus help reduce the cost and radiation dose in the workup of at-risk patients for CVD with quantitative information from a single LDCT exam. Given the technical challenges associated with the quantification of CAC from LDCT for lung cancer screening versus ECG-gated CCT, our study indicates a significant development in establishing a CVD-related risk nomogram with LDCT.

To interpret the prediction results of Tri2D-Net, we generated heatmaps using the Gradient-weighted Class Activation Mapping (Grad-CAM[37]) and exported the attention maps from the attention block. Figure 5 shows the results of three representative subjects from the NLST dataset. Figure 5 a, b belong to two subjects who died of CVD, referred as Case a and Case b, respectively. Figure 5c shows the image of a subject case c, who survived by the end of the trial. Case a has severe CAC with an average CAC Grade of 3.0 by the three radiologists. Tri2D-Net

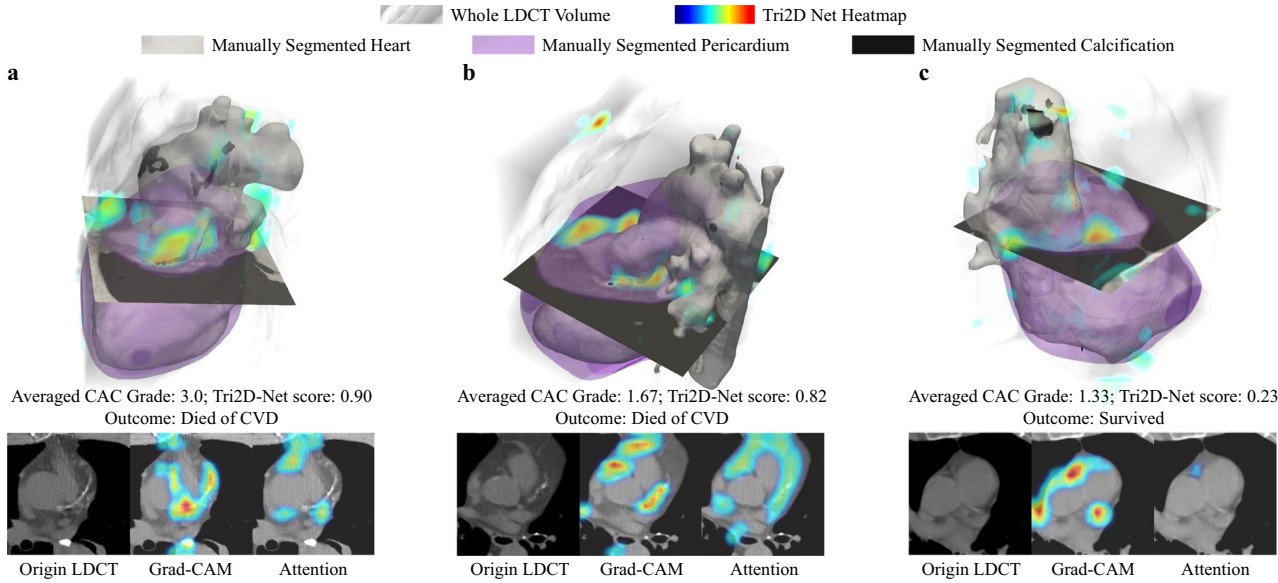

**Fig. 5 Visualizations of the features learned by the Tri2D-Net.** The Gradient-weighted Class Activation Mapping (Grad-CAM[37]), and the attention maps of Case (**a**), Case (**b**), and Case (**c**) learned by the attention block in the Tri2D-Net are visualized with heatmaps. The Grad-CAM[37] is a widely used visualization technique for CNN that produces a coarse localization map highlighting the important regions in the image for the final prediction.

captured the strong calcium as shown in the Grad-CAM heatmap and predicted a high CVD risk score of 0.90. Case b (Fig. 5b) had mild to moderate CAC with an average CAC Grade of 1.67. However, the attention block noticed abundant juxtacardiac fat and Tri2D-Net gave a high score of 0.82 for the case. Case c has mild CAC graded as 1.33 by the radiologists. Since there was little calcification and juxtacardiac fat as indicated by the heatmaps, Tri2D-Net predicted a low-risk score of 0.23 for this survived patient. Visualization of these cases demonstrates the contributions of both CAC and juxtacardiac fat to our model for predicting CVD risk scores in contrast with the mere reliance on CAC as the sole biomarker in prior studies[17,28–31]. It is consistent with the clinical findings that epicardial/pericardial fat correlates with several CVD risks[38,39]. Based on this finding, we further examined the Pearson correlation between our model-predicted CVD risk score and juxtacardiac fat volume. To measure the volume of juxtacardiac fat, we first used an existing deep learning based heart segmentation mode[17] to segment the heart region. Then, we identified fat inside the segmented heart by HU-based cutoffs (voxels in [-190,-30] HU were considered to represent fat). Our model achieved a Pearson correlation of 0.199 ($p < 0.0001$) with juxtacardiac fat volume. In contrast, the radiologist estimated CAC and the deep learning estimated CAC[17] got Pearson correlation of 0.078 ($p = 0.0004$) and 0.085 ($p = 0.0001$), respectively. The ability to capture various features in addition to CAC makes our model superior to the existing CAC scoring models for CVD screening and CVD mortality quantification.

Our study's limitation includes using CVD-related ICD-10 codes for labeling the subjects, which may miss some CVD-related deaths or mislabel patients who died from other heart diseases as CVD mortality. Mitigating its influence on our study motivated us to collect data at MGH and evaluate our model using the surrogate gold-standard CVD risk scores on the MGH dataset. Our results on the MGH dataset support the pretrained model's utilities on data from a different source. Another limitation is that we did not have access to a sufficient number of patients with mortality information from CVD. The use of LDCT at MGH (as well as from other United States sites) started after the United States Preventive Services Taskforce recommended

annual screening for lung cancer with LDCT in 2013. We believe that with the increasing use of LDCT over time, CT data with CVD mortality information will become available. It is worth noting that the proposed method demonstrated on this specific application may apply to other clinical studies, for example, predicting the cause of death among pediatric patients with cancer[40]. There are reports on increased heart disease-related fatalities of cancer patients in general, likely related to decreased cancer-specific mortality and aging population[41]. Older age-group, male gender, African American race, and unmarried status are key risk factors for cardiac mortality in cancer patients. Another study based on data from the Surveillance, Epidemiology, and End Results program reported higher risk of fatal stroke in patients with cancer[42]. The study may help reduce healthcare disparity to benefit those high-risk patients with disadvantaged socioeconomic status.

## Methods

**Development and validation datasets.** The NLST dataset was used for the deep learning model development. In each exam of NLST, multiple CT volumes were generated with different reconstruction kernels. We used 33,413 CT volumes from 10,395 subjects (for more details, see Supplementary Fig. 2). Those subjects were randomly divided into training (7268, 70%), validation (1042, 10%) and test sets (2085, 20%). The participants were enrolled in the trial from August 2002 through April 2004. Subject health history, reports of the annual LDCT screening scan and death information were collected through December 31, 2009. All participants enrolling in NLST signed an informed consent developed and approved by the IRBs at each of the 33 participating medical institutions, the National Cancer Institute (NCI) IRB, and the Westat IRB. The NLST data are publicly available through the Cancer Data Access System (CDAS) of the National Institutes of Health (NIH). LDCTs were collected from multiple institutions, with slice spacing varying from 0.5 to 5 mm. Scans with slice spacing larger than 3mm or with scan length along superior to inferior less than 200 mm were filtered out. Supplementary Fig. 2 shows the inclusion and exclusion criteria. Since the NLST was designed for lung cancer screening, CVD-related information is incomplete. Therefore, we only used LDCT images with clear CVD information. Specifically, for all CVD-negative patients and CVD-positive patients who died in the trial with CVD-related causes, all available LDCT exams were considered as valid exams. For other CVD-positive patients, only the exams with clear CVD abnormalities reported were considered as valid exams. In the training set, all LDCT volumes generated in the valid exams were treated as independent cases to increase the number of training samples. In the validation and testing phases, a single CT volume was randomly selected from the valid exams of each subject to preserve the original data distribution. Since the number of LDCT exams is inconsistent across subjects (for example, patients with

death or diagnosis of lung cancer on initial LDCTs did not complete all the follow-up LDCTs), the use of these CT volumes as independent cases can change the data distribution and introduce a bias to the results. After the random selection, the formed test set keeps an averaged follow-up time of 6.5 ± 0.6 years. Other properties of the NLST dataset are summarized in Tables 1, 2. More detailed information including manufacturer, scanner, and reconstruction kernel can be found in Supplementary Table 2.

The MGH dataset was collected at Massachusetts General Hospital (MGH at Boston, MA) in 2019. The retrospective study was approved by IRB with the waiver of informed consent. It was in compliance with the Health Insurance Portability and Accountability Act (HIPAA). By reviewing the electronic medical records (EPIC, Epic Systems Corporation) at MGH, 348 adult patients were identified, who had clinically indicated LDCT for lung cancer screening within a 12-month period. 248 of them also had ECG-gated CT angiography and coronary calcium scoring within a 12-month period. Thirteen subjects were excluded because they had coronary stents, prosthetic heart valves, prior coronary artery bypass graft surgery, or metal artifacts in the region of cardiac silhouette. The final dataset contains 335 adult patients with details in Table 1. The data collected from each participant contains one chest LDCT image, one non-contrast ECG-gated cardiac CT (CCT) image, a CAC score (Agatston score)[34], a Coronary Artery Disease Reporting and Data Systems (CAD-RADS$^{TM}$) score[35] semi-automatically calculated from the CCT image, and a MESA 10-year risk score (MESA score)[36] calculated with the standard clinical protocol. Dicom files were handled with the free dicom viewer MicroDicom DICOM Viewer V3.01 (https://www.microdicom.com/). LDCTs were reconstructed at 1-1.25 mm section thickness at 0.8–1 mm section interval using vendor-specific iterative reconstruction techniques. More detailed information including manufacturer, scanner, and reconstruction kernel can be found in Supplementary Table 3. It is noteworthy that the MGH dataset was not used for training or fine-tuning our proposed network, but only for evaluating/testing the performance of the deep learning model on LDCT to compare with human experts defined standards from CCT.

**Model development**. Challenges of chest LDCT-based CVD risk estimation mainly come from three aspects. First, while a chest LDCT image volume has a large field view, the heart takes only a small subset. Features extracted by a deep learning model on the entire image may hide the CVD risk related information[33]. To tackle this problem, a cardiac region extractor was developed to locate and isolate the heart so that the risk estimation model can focus on this region of interest. Second, chest LDCT images are inherently 3D. In deep learning, 3D convolution neural networks (3D-CNNs) are needed for 3D feature extraction. However, popular 3D-CNN methods are either hard to train because of a huge amount of parameters[43] like C3D[44], or need pretrained 2D-CNNs for parameter initialization like I3D[43]. At the same time, radiologists typically view CT images in 2D using the three orthogonal views: axial, sagittal, and coronal planes. Therefore, we designed a Tri2D-Net to efficiently extract features of 3D images in three orthogonal 2D views. The key details of the networks are presented as follows. Third, the risk of CVD is hard to quantify. It could contain various diseases and symptoms. We combine information from various medical reports, including CT abnormalities reports, causes of death, and CVD histories, in NLST dataset and transfer the risk estimation task into a classification task. The whole pipeline of the proposed method is shown in Supplementary Fig. 1.

- *Heart detector:* RetinaNet[45] is used in our work to detect the heart region of interest. To train this network, we randomly selected 286 LDCT volumes from different subjects in the training set. We annotated the bounding boxes of the cardiac region including the whole heart and aorta slice by slice in the axial view. In application, the trained detector is applied to each axial slice for independent localization. Then, the two extreme corner points $A(x_{min}, y_{min}, z_{min})$ and $B(x_{max}, y_{max}, z_{max})$ are identified by finding the maximal and minimal coordinates in all the detected bounding boxes, which defines the region enclosing the heart and excluding most of the irrelevant anatomical structures.

- *CVD risk prediction model:* As shown in Supplementary Fig. 1, the proposed Tri2D-Net consists of two stages, feature extraction, and feature fusion. The former uses three 2D CNN branches to independently extract features from the three orthogonal views. By setting a cross-entropy loss for each branch, the three 2D CNN branches separately receive direct feedback to learn their parameters. This leads to a dramatically reduced optimization space compared to a massive 3D network[46]. Specifically, in each branch, we split the original Resnet-18[47] into two parts, the first 13 layers (L+13) and the last 5 layers (L-5). To introduce clinical prior knowledge into the network, we applied an attention block to help the model focus on calcification and fat regions, which highly correlate with CVD[48]. The attention block first selects the calcification and fat regions with HU ranges (calcification: HU > 130; fat: HU in [ − 190, − 30]). Then the masked slides are separately fed into a 4 layers CNN (first 4 layers of VGG11[49]) to generate an attention map for each slide. The feature fusion module then concatenates the three feature representations extracted by the 2D CNN branches and feeds the result to a classifier for the final prediction.

**Statistical analysis**. All confidence intervals of AUC values were computed based on the method proposed by Hanley and McNeil[50]. p values of significance test on AUC comparison were calculated using the z-test described by Zhou et al.[51]. p values for sensitivity comparisons were calculated through a standard permutation test[52] with 10,000 random resamplings.

**Reporting summary**. Further information on research design is available in the Nature Research Reporting Summary linked to this article.

## Data availability

This study used the NLST dataset, which is publicly available at https://biometry.nci.nih.gov/cdas/learn/nlst/images/. The MGH dataset was not part of the NLST dataset. The deidentified dataset from MGH was used with an institutional review board-approval for the current study. The data sharing regulations at MGH forbid open access to its patients' data. Any access to deidentified MGH data, or a test subset, requires IRB and MGH Data Sharing Committee approvals both at the MGH and at the requesting institution (details of how to request access are available from Dr Mannudeep Kalra at MGH). Source data are provided with this paper.

## Code availability

The code used for training the models and the parameters of the pretrained deep learning networks has been made publicly available[53] at https://github.com/DIAL-RPI/CVD-Risk-Estimator/. We have also packaged our model into an open-access and ready-to-use tool (https://colab.research.google.com/github/DIAL-RPI/CVD-Risk-Estimator/blob/develop/colab_run.ipynb) for the community to test and feedback.

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

## Acknowledgements

This work was partly supported by National Heart, Lung, and Blood Institute (NHLBI) of the National Institutes of Health (NIH) under award R56HL145172. The authors thank the National Cancer Institute (NCI) for access to NCI's data collected by the National Lung Screening Trial. The statements contained herein are solely those of the authors and do not represent or imply concurrence or endorsement by NCI.

## Author contributions

G.W., M.K.K., and P.Y. initiated and supervised the project, and provided the concept and design of the experiments. H.C., H.S., G.W., and P.Y. developed the network architecture and analyzed the data. H.C. trained the network and reported the results with figures and tables. H.S. contributed machine learning expertise in network design and CT image analysis. F.H., R.S., R.D.K., and M.K.K. provided clinical expertise, acquired and annotated the MGH dataset, and screened NLST LDCT images. H.G. advised on data processing and performed additional experiments of a benchmark method. T.C. contributed on data curation. H.C. H.S., G.W., M.K.K., and P.Y. wrote the manuscript. All the authors reviewed and revised the manuscript.

## Competing interests

The authors declare no competing interests.

**Additional information**

