## [Peer Review File · Nature Communications]

Reviewers' Comments:

Reviewer #1:

Remarks to the Author:

The authors use data from low dose CTs (LDCTs) for lung cancer screening to extract data on risk of cardiovascular disease (CVD). Their deep model was trained with 30,286 LDCT volumes and achieved the "state-of-the-art performance" (area under the curve (AUC) of 0.869) on 2,085 National Lung Cancer Screening Trial (NLST) subjects, and effectively identified patients with high CVD mortality risks (AUC of 0.768). The deep model was further calibrated against the clinical gold standard CVD risk scores from 3 other scores from an independent dataset of 106 subjects.

This is an important field of study, and it would be helpful to identify patients who are at risk to die of CVD from LDCTs. Indeed, this population of patients is at risk to die of both causes of death, lung cancer and heart disease. The results seem noteworthy, based on the AUCs and the ability to predict death in the validation sets. This would be a significant work to this field and related fields, pending addressing of the major comments below.

I have two major comments:

(1) I am unclear on the endpoint: Is overall survival an outcome, death from heart disease, death from lung cancer, or death from other causes? In clinical practice, it would be much more important to show that data from LDCTs may be used to predict death from heart disease, or any cause, rather than to say that data from LDCT can predict another surrogate score (eg, CAD-RADS, MESA, etc). If mortality is presented, I would include Kaplan Meier curves and cumulative incidence plots of death from any cause, heart disease specific mortality, and lung cancer specific mortality.

LDCT screening is very controversial because it does not appear to decrease overall mortality. Per the NELSON trial of LDCT (NEJM, 2020; PMID: 31995683), overall mortality rates are similar between the LDCT screening group and non-screening group (though lung cancer specific mortality is slightly decreased in LDCT group), so I wonder if the patients in this LDCT study are dying of heart disease vs lung cancer vs other causes. The causes of death would be important to point out in the current work.

Additionally, the US NLST (NEJM, 2011) showed that the number needed to screen to prevent one lung cancer death was 320. In the current study, what would be the NNS to prevent one death from heart disease or other cause?

Along the same lines, how many patients died of unknown causes? For patients with heart disease, it is not uncommon for a patient to collapse or be found dead at home. ICD-9 coded causes of death are tricky in this instance, because the patient may have died from MI, PE, stroke, etc. If the patient has a history of a lung cancer (even if it is a T1 N0 M0 lesion), the lung cancer may be implicated in the patient's death (even though the cancer very likely did not kill the patient).

(2) For many clinicians, myself included, machine learning is a black box, and it is one that clinicians can never use. There is the stereotypical figure (Figure 1A/1B) of data going in, "learned features" and then some sort of risk calculated (Figure 1C). How does the machine actually do this? Is machine learning superior to other methods, eg logistic regression, random forest? Can the authors provide an example of a CT (like a video) where the machine performs this calculation? And could readers use this work? It would be great if a clinician like myself could pull up a CT of the chest through PACS, open it in the authors' program (or at least provide key slices for the program), and the program could provide a risk score of death from heart disease vs other causes. I think that for clinicians to use this in practice is the ultimate goal. With "standard" logistic regression, multivariate analysis, and nomogram, this could be estimated. Currently, as a reader of this work, it appears the authors have a good model, but it is unclear how it works, and I don't know how I can apply it to my own practice (though I would really like to).

Thus, I cannot comment on the methods. I would recommend a machine learning expert review this work, because I am not familiar with their methods and comparison to KAMP-Net or AE+SVM.

Other comments:

- I would discuss more about the risk of death from competing causes in the discussion section. For example, the Nature Comms journal has published the following on this topic: PMID 32332714, 31729378
- CVD is an overarching term that includes heart disease, stroke, aneurysm, etc. I recommend the authors just talk about heart disease.
- Could the authors please discuss how these CTs could be used to follow pediatric cancer patients to predict competing death? For example, PMID 32298481
- The grammar / syntax needs to be corrected in a few areas, eg, "and achieved the state-of-the-art performance"
- Also, state of art performance is very subjective. I would delete this.
- There are references in the abstract, and this cannot be present.

Reviewer #2:

Remarks to the Author:

In this paper, the authors present a deep learning model for CVD risk analysis from LDCT collected in a lung cancer screening setting. The deep learning method extracts the heart area from the CT scan, and performs an analysis of that area to produce a binary score whether signs of CVD are present or not. A subject was considered CVD-positive if any cardiovascular abnormality was reported in the subject's CT screening exams or the subject died of CVD. A CVD-negative subject has no CVD related medical history and no reported cardiovascular abnormality in any of the CT scans during the trial, and did not die of circulatory system diseases. The authors trained their system using data from the NLST trial, for which they also set 20%, a set of 2085 subjects, aside for testing. Finally, a retrospective dataset of 119 subjects is collected from MGH and the performance of the model is evaluated against the three clinical standards for measuring CVD from cardiac CT: CAC scores, CAD-RADS scores, and MESA scores.

Major points:

- The paper is unclear about what the deep learning model precisely predicts. Based on my review of the paper, I think the model is primarily intended to predict whether there are signs of CVD or not. However, this is quite vague in the abstract and needs to be clarified. This model is not directly trained for mortality prediction, I think. However, page 3 reads "A deep CNN model for CVD mortality risk analysis of LDCT images was designed and trained on the NLST dataset". Further on in this paragraph, it reads that the model was trained for CVD positive or CVD negative so this is unclear and really needs clarification. If the model intends to predict mortality, it needs a time frame needs to be attached to that. Mortality within five years, one year? 10 years?
- The external validation is limited - a relatively small dataset of 119 patients is retrospectively collected. In addition, the authors retrained a logistic regression model using 5-fold CV using the features from the deep learning model? Why? In this way, it is not a direct external validation of the model developed on the NLST data, but more a validation of the feature extractor part of the DL model. Also, it is not completely clear what features are used. The output of the fully connected layer? How many features are there in this feature representation? Since a five-fold CV is performed, are the selected features for the logistic regression stable?
- For the NLST cases, what CT scan is used? This is unclear. If all CT volumes are used, how do the authors get one score per patient? Is the maximum deep learning score across the different volumes used? Why not perform an analysis per CT when assessing the performance of the model to detect signs of CVD?
- The abstract and introduction are too strong with regards to the benefits of using LDCT for CVD screening. At present, there is no scientific consensus whether the benefits of LDCT screening for CVD outweigh the harms. LDCT screening has only shown benefit for reducing lung cancer

mortality in a specific high-risk population. The introduction and abstract should be changed to reflect this and explain the current situation better.

Minor issues:

- NLST is quite old data. What effect does this have on the model?
- The claims in the Discussion need to be toned down. No prospective data yet, so this is only preliminary evidence.
- The deep learning model mostly focuses on the CAC, it seems, as the AUC is 0.942 for this, while lower for MESA and CAD-RADS (0.817 and 0.809 respectively). Isn't this method primarily picking up the calcifications in the heart? Any analysis done to check this?
- Since reader group 2, consisting of only one reader, performed better, please split out the experience levels of the three readers (M.K.K., R.S. and R.D.K.). Is there a difference in experience level?
- The authors write "Then, a max pooling is employed on the slide dimension to fuse all these feature maps into one feature map". So, Resnet-18 has 256 feature maps after L13. The feature map size is $14 \times 14 \times 256 \times z$, where z is the number of slices. Then, a max pool is done over the z -direction, which leads to a new feature map of $14 \times 14 \times 256$? If yes, this sentence should be rephrased because it reads 'one feature map', which is confusing.
- How were the heatmaps generated? This needs to be clarified.

Reviewer #3:

Remarks to the Author:

In the manuscript, the authors developed a deep learning algorithm to predict the cardiovascular risk of a subject from a low dose chest CT for lung cancer screening. The deep learning algorithm was trained with the NLST dataset, and exhibited radiologist-level performance in detection of cardiovascular disease and prediction of cardiovascular mortality in a split-sample test dataset. In an independent test dataset, the algorithm also exhibited nice performances for prediction of cardiovascular risk scores obtained from ECG-gated cardiac CTs.

I believe that the result is meaningful, because cardiovascular disease is a major source of mortality in population who undergo lung cancer screening and identifying high risk patients from the low dose chest CT using deep learning technique without additional examination may help patients to receive appropriate intervention.

Below are my specific comments and suggestions regarding the manuscript:

1. The NLST dataset were labeled as normal or abnormal based on LDCT report, medical history, and cause of mortality. Since cardiovascular abnormalities on LDCT or cardiovascular diseases have wide spectrum of severity or clinical relevance (e.g. from tiny coronary calcification to overt sign of heart failure, and from mild arterial hypertension to myocardial infarction), the authors need to provide clear definition of abnormal label, or actual spectrum of LDCT abnormalities or cardiovascular diseases in the NLST dataset.
2. Regarding the interpretation by radiologists, radiologists focused on the presence and extent of coronary artery calcifications, for prediction of cardiovascular disease or related mortality. However, there may other abnormalities suggesting cardiovascular diseases such as enlarged heart chamber, dilatation or calcification of aorta, and pleural or pericardial effusion, and the performance of radiologists for identification of cardiovascular diseases may have been underestimated.
3. The authors may provide representative cases along with the heat maps from the algorithm, to demonstrate whether the algorithm actually focused on findings of cardiovascular diseases (e.g. coronary calcification).
4. For cardiovascular mortality prediction in the NLST dataset, performance of radiologists exhibited substantial difference. I wonder whether there was difference in the experience of radiologists and less-experienced radiologist exhibited lower performance.
5. In the MGH dataset, the algorithm was evaluated against various cardiovascular risk scores from the ECT-gated cardiac CTs, but not evaluated against actual presence of cardiovascular

disease or related mortality. I believe that evaluation against mortality might be practically impossible, however, performance against the presence of cardiovascular disease (as in the NLST) may help readers to understand the true performance of algorithm in an independent dataset.

6. Cardiac motion artifact and noise are major cause of limited evaluation of coronary artery calcification in the LDCTs, and their degrees are directly related with scanning and reconstruction protocols. Therefore, the authors may present detailed scanning and reconstruction protocols (e.g. CT scanner, pitch, reconstruction kernel, ...) of LDCTs (especially in the MGH dataset).

7. Although I am not a statistical expert, I am concerned about whether the ROC analyses for cardiovascular mortality ignoring the time to death and censoring is appropriate.

Responses to Reviewers' Critiques

NCOMMS-20-33634-T

**Deep Learning Predicts Cardiovascular Disease Risks from Lung
Cancer Screening Low-dose Computed Tomography**

Hanqing Chao, Hongming Shan, Fatemeh Homayounieh, Ramandeep Singh,
Ruhani Doda Khera, Hengtao Guo, Timothy Su,
Ge Wang* , Mannudeep K. Kalra*, Pingkun Yan*

*Co-corresponding author e-mails:

wangg6@rpi.edu,

MKALRA@mgh.harvard.edu

yanp2@rpi.edu

Comments and Responses

***AE Comment:** “As you will see from the reports copied below, the reviewers raise important concerns. We find that these concerns limit the strength of the study, and therefore we ask you to address them with additional work. Without substantial revisions, we will be unlikely to send the paper back to review. In particular, adding additional external validation data would increase chances for publication.”*

Response: We are highly grateful to the editor and the reviewers for assessing our manuscript and providing constructive critiques, which have significantly improved our research quality. We believe that we have fully addressed all the comments with major revisions to our work, as detailed in our point-to-point responses below. The manuscript has also been revised accordingly, and the changes are highlighted in blue. In summary, we have made the following four key revisions:

- 1. Augmentation of the validation datasets and additional experiments with enhanced results:** We have tripled the size of the independent CVD screening dataset acquired at MGH from 106 patients to 335 patients. As recommended, we have performed a direct external validation of the deep learning model on CVD screening. On the NLST dataset, as suggested, three radiologists have independently annotated all the test studies including calcium levels of 2,085 CT scans. All the relevant experiments have been re-performed. The new findings are consistent with our previous results, demonstrating the potential of LDCT for dual screening of lung cancer and CVD.
- 2. Attention in the network architecture for interpretability:** To interpret the performance of our network, we have improved the network architecture to extract image features with a novel attention mechanism. Also, we have visualized the feature extraction workflow, showing important network behaviors.
- 3. Transparency and reproducibility of our network:** To improve the transparency and reproducibility of our work, we have publicly released our source code on GitHub (<https://github.com/DIAL-RPI/CVD-Risk-Estimator/>). We have also packaged our model into an open-access and ready-to-use tool (https://colab.research.google.com/github/DIAL-RPI/CVD-Risk-Estimator/blob/develop/colab_run.ipynb) for the community to test and feedback.
- 4. First CVD nomogram from deep features (CAC, pericardial fat, and others):** We would like to emphasize that our work represents a significant advancement over current estimation of CAC, which is already a function of some commercial cardiac CT software. Beyond CAC computation, we offer the first nomogram from CAC, pericardial fat, and other deep features to predict patient CVD mortality risks. In addition to our deep learning approach's competitive performance relative to radiologists' CAC grading of lung cancer screening LDCT images, the developed model computed and highly correlated with three clinical gold standard scores on our independently collected ECG-gated cardiac CT images and obtained exciting results.

Response to Reviewer#1:

The authors use data from low dose CTs (LDCTs) for lung cancer screening to extract data on risk of cardiovascular disease (CVD). Their deep model was trained with 30,286 LDCT volumes and achieved the “state-of-the-art performance” (area under the curve (AUC) of 0.869) on 2,085 National Lung Cancer Screening Trial (NLST) subjects, and effectively identified patients with high CVD mortality risks (AUC of 0.768). The deep model was further calibrated against the clinical gold standard CVD risk scores from 3 other scores from an independent dataset of 106 subjects.

This is an important field of study, and it would be helpful to identify patients who are at risk to die of CVD from LDCTs. Indeed, this population of patients is at risk to die of both causes of death, lung cancer and heart disease. The results seem noteworthy, based on the AUCs and the ability to predict death in the validation sets. This would be a significant work to this field and related fields, pending addressing of the major comments below.

• **Major Comments**

I have two major comments: (To improve the clarity of our response, we have split each major comment into a set of sub-comments for responding.)

1) Comment #1:

A) I am unclear on the endpoint: Is overall survival an outcome, death from heart disease, death from lung cancer, or death from other causes?

Response: There are two endpoints in this project: i) diagnosed cardiovascular diseases (CVDs), and ii) CVD related mortality. For clarification, we’ve revised the Introduction section (Page 2, Paragraph 3) as follows:

*“To tackle the limitations of the prior studies, we built an end-to-end deep neural network to **a)** screen patients for CVDs and **b)** quantify CVD mortality risk scores directly from chest LDCT examinations. Specifically, our approach focuses on the cardiac region in a chest LDCT scan and makes predictions based on automatically learned comprehensive features of CVDs and mortality risks. The prediction is calibrated against the incidence of CVD abnormalities during the follow-up period of a clinical trial, subjective assessment of radiologists in reader studies, and the CVD risk scores calculated from electrocardiogram (ECG)-gated cardiac CT including the CAC score³³, CAD-RADS score³⁴, and MESA 10-year CHD risk score³⁵.”*

B) In clinical practice, it would be much more important to show that data from LDCTs may be used to predict death from heart disease, or any cause, rather than to say that data from LDCT can predict another surrogate score (eg, CAD-RADS, MESA, etc).

Response: We agree! We performed the experiment of mortality prediction on the NLST dataset in our initial manuscript. It would be ideal if we could directly evaluate the proposed algorithm on the MGH dataset for CVD mortality prediction. However, since the MGH dataset was collected from a patient cohort after 2015, the mortality records are not available for CVD mortality prediction. Therefore, we could only use the clinically adopted risk scores – derived from ECG-gated cardiac CT scans – as our best surrogates to evaluate the trained model for mortality prediction. To compensate for this limitation, in the revised manuscript, we have expanded our MGH dataset from 106 patients to 335 patients for evaluating CVD screening. Of those, 235 patients, who underwent ECG-gated cardiac CT exams, were further used to

evaluate the estimated risk scores. In the revision, the detailed description of the expanded MGH dataset can be found on Page 4, Paragraph 4:

“Furthermore, through an institutional review board (IRB) approved retrospective study, we acquired an independent and fully de-identified dataset from MGH in 2020. This MGH dataset contains 335 patients (161 men, 174 women, mean age 63.6 ± 8.0 years), who underwent LDCT for lung cancer screening. Of these, 100 patients had no observed CVD abnormalities in their LDCT. The remaining 235 subjects underwent ECG-gated cardiac CT for CVD risk assessment for atypical chest pain, equivocal stress test, and chest pain with low to intermediate risk for CHD. Three CVD risk scores were calculated for the 235 subjects from their cardiac CT images, including CAC score³³, coronary stenosis (quantified as CAD-RADS)³⁴, and MESA 10-year CHD risk³⁵. Table 1 lists the characteristics of the dataset (see Methods, MGH dataset). The MGH dataset was used to evaluate the clinical significance of the NLST-trained model for feature extraction without re-training or fine-tuning. For the validation of CVD screening, a subject was labeled as CVD-positive if the subject underwent an ECG-gated cardiac CT screening and received either a CAC score > 10 or a CAD-RADS score > 1 . Correspondingly, a CVD-negative subject had either all the LDCT screening exams being negative or the calculated scores of $CAC \leq 10$ and $CAD-RADS \leq 1$. Based on the above criteria, 181 subjects were CVD positive and 154 patients were CVD negative. For the quantification of CVD mortality risk, since this MGH dataset was collected from a patient cohort, who had recent exams (2015-2020), the mortality records are not yet available. We instead calibrate our model against the three gold standard risk scores as surrogate evaluators.”

The corresponding results and analysis on the newly added CVD screening experiment are at Page 7, Paragraph 3 and Fig. 4a:

“In the experiment of CVD screening shown in Fig. 4a, our deep learning model achieved a significantly higher ($p < 0.0001$) AUC value of 0.924 (95% confidence interval, 0.909-0.940) than its performance on the NLST dataset (0.871), where the network was originally trained. Superior performance on this external MGH dataset may be due to the better image quality of the contemporary MGH dataset acquired from newer scanners. Furthermore, the MGH annotated datasets and use of ECG-gated cardiac CT as a gold standard may have been superior to the annotation in NLST. This experiment shows that the proposed deep learning model has good generalizability and is highly consistent with human experts' joint label for CVD screening from LDCT and CVD scores calculated on ECG-gated cardiac CT.”

Results and analysis of the surrogate CVD risk score prediction on the expanded MGH dataset are summarized from Page 9 to Page 10 and Fig. 4b, c, & d:

“The features extracted by the trained CVD feature extractor from LDCT were first used to estimate the CAC score³³. With a threshold of 400 for CAC scores, the MGH subjects were divided into two groups: 78 subjects with severe CAC and 157 subjects with none or minor CAC. Our model achieved an AUC value of 0.942 (95% confidence interval, 0.927-0.958) and significantly outperformed the other two methods ($p < 0.0001$, see Fig. 4b). It is worth noting that our model is competitive with experienced radiologists. The AUC of our DL model was only slightly lower than the radiologists' CAC grading without significant difference ($p > 0.43$), despite the fact that our DL model has never been trained for CAC score estimation. These results suggest that deep learning analysis of LDCT can well approximate the human expert performance using CCT in differentiating patients with severe and non-severe CAC.”

Figure 4: **Results of 4 experiments on the MGH dataset. a**, Validation of CVD screening. **b & c & d**, Comparison of our deep learning model with three clinical standard criteria the CAC score³³ (**b**), CAD-RADS³⁴ (**c**), and MESA 10-year risk score³⁵ (**d**) calculated with the standard protocol. For radiologists' CAC grades in **b**, red dots represents CAC Grade 1+, yellow dots represents CAC Grade 2+, green dots represents CAC Grade 3.

The second experiment evaluates the capability of the deep learning model in classifying subjects into high and low risk groups using LDCT by comparing against the coronary stenosis (CAD-RADS) scores³⁴ obtained by human experts on CCT. Subjects with CAD-RADS scores greater than or equal to 4 are labeled as with severe stenosis, i.e., positive samples (51 subjects). The other 184 subjects with smaller scores were labeled as negative. Our model reached an AUC value of 0.808 (95% confidence interval, 0.758-0.858, see Fig. 4c). Our model significantly outperformed the other two methods ($p \leq 0.0338$). Unlike calcification, coronary stenosis is much harder to detect through a chest LDCT screening, while it is a direct biomarker of CVD risk. The performance obtained using LDCT is thus highly encouraging.

The superiority demonstrates that our model can quantify the subclinical imaging markers on LDCT, making it a promising tool for CVD assessment in lung cancer screening.

In the third experiment, patients were divided into high and low risk groups according to MESA 10-year risk score³⁵, which is a clinical gold-standard risk stratification score for CVD integrating multiple factors including gender, age, race, smoking habit, family history, diabetes, lipid lowering and hypertension medication, CAC score extracted from CCT, and laboratory findings including cholesterol and blood pressure. Because some of the 235 subjects did not have all the needed exams, we are only able to calculate the MESA scores of 106 subjects. When median MESA 10-year risk score in our patients was used as a threshold (14.2), 52 subjects with greater scores were labeled as high risk, while the other 54 subjects were labeled as low risk. Our model achieved an AUC value of 0.799 (95% confidence interval, 0.736-0.863), which significantly outperformed all the other methods (see Fig. 4d).”

C) If mortality is presented, I would include Kaplan Meier curves and cumulative incidence plots of death from any cause, heart disease specific mortality, and lung cancer specific mortality.

Response: Thanks for the suggestion! We have added a Kaplan Meier curve for the NLST dataset on CVD related mortality (Fig. 3). The corresponding analysis of these results can be found as follows (Page 7, Paragraph 1):

“For further comparison, Fig. 3 shows the Kaplan Meier curves of different risk groups labeled by our model and the radiologists, respectively. For the radiologists, we used the average reader prediction of CAC Grade 2+ to separate the subjects into low/high risk groups and drew the Kaplan Meier curves for both groups. The final survival probabilities of low and high risk groups by radiologists are 95.79% and 85.83%, respectively. For fair and direct comparison, in Fig. 3a, we selected a threshold to divide the quantified CVD risk scores using our model so that the low risk group has a survival probability of 95.79%, similar to the radiologists. Under this circumstance, the model-predicted high-risk group showed a significantly lower ($p=0.0059$) final survival probability of 73.24%. Similarly, in Fig. 3b, we selected a threshold so that the high-risk group has a survival probability of 85.75%, also similar to the radiologists. In this case, the model-predicted low-risk group achieved a significantly higher ($p=0.0272$) final survival probability of 97.46%.”

Figure 3: **Kaplan Meier curves on the NLST dataset.** Comparison of our model with the radiologists. Thresholds were selected on the CVD risk score calculated by our model to enforce the low (a) or high (b) risk group has a similar survival probability with that of the radiologists.

D) LDCT screening is very controversial because it does not appear to decrease overall mortality. Per the NELSON trial of LDCT (NEJM, 2020; PMID: 31995683), overall mortality rates are similar between the LDCT screening group and non-screening group (though lung cancer specific mortality is slightly decreased in LDCT group), so I wonder if the patients in this LDCT study are dying of heart disease vs lung cancer vs other causes. The causes of death would be important to point out in the current work.

Response: Great point! We agree with the reviewer that using LDCT for lung cancer screening only does not seem to decrease the overall mortality as shown by the literature. Heart diseases, the leading cause of death for patients in NLST, should also be screened. The observation motivated our study, which tries to reuse the LDCT scans of those patients to conduct a “radiation free” CVD risk screening. We have incorporated the reviewer’s comments into our manuscript as follows (Page 1 Paragraph 1):

“Cardiovascular disease (CVD) affects nearly half of American adults and causes more than 30% offatality¹. The prediction of CVD risk is fundamental to the clinical practice in managing patienthealth². Recent studies have shown that the patients diagnosed with cancer have a ten-fold greater risk of CVD mortality than the general population³. For lung cancer screening, low dose computed tomography (LDCT) has been proven effective through clinical trials^{4, 5}. In the National LungCancer Screening Trial (NLST), 356 participants who underwent LDCT died of lung cancer during the 6-year follow-up period. However, more patients, 486 others, died ofCVD. The NELSON trial shows similar overall mortality rates between the study groups even though the lung cancer mortality decreased in the LDCT screening group⁶. Therefore, screening significant comorbidities like CVD in high-risk subjects undergoing LDCT for lung cancer screening is critical to lower overall mortality. Nevertheless, when the cancer risk population receives cancer screening, radiologists overlook their potential CVD risk.”

E) Additionally, the US NLST (NEJM, 2011) showed that the number needed to screen to prevent one lung cancer death was 320. In the current study, what would be the NNS to prevent one death from heart disease or other cause?

Response: In NLST, the NNS was calculated by comparing the actual deaths in the LDCT group and the chest X-ray group. However, our study aims to analyze existing LDCT exams for CVD screening, which is not a clinical trial and thus has no control group to compare with. To evaluate the clinical importance of our work, in our revised manuscript, we report a sensitivity on the death prediction with the control of positive predictive value (PPV) on Page 6 Paragraph 2. We then compared the prediction results of our model with those of the radiologists through a reader study. The corresponding content in the paper has been revised as follows:

“With the same PPV of averaged CAC Grade 2+ (10.8%), our model achieved a sensitivity of 80.8%. Specifically, in the NLST test set, our model successfully identified 97 of the 120 deceased subjects as high risk, while the averaged CAC Grade 2+ labeled 35 of those 97 cases as low risk.”

F) Along the same lines, how many patients died of unknown causes? For patients with heart disease, it is not uncommon for a patient to collapse or be found dead at home. ICD-9 coded causes of death are tricky in this instance, because the patient may have died from MI, PE, stroke, etc. If the patient has a history of a lung cancer (even if it is a T1 N0 M0 lesion), the lung cancer may be implicated in the patient’s death (even though the cancer very likely did not kill the patient).

Response: Once again, we agree with the reviewer. Indeed, the ICD-10 codes may be imperfect for our study. We have added this as a limitation of our work (Page 11 Paragraph 2). Mitigating its influence on our study motivated us to collect new data at MGH and evaluate our model using the surrogate gold standard CVD risk scores on the MGH dataset. Our results on the MGH data support the utilities of our trained model.

“Our study’s limitation includes using ICD-10 codes for assigning the cause of deaths as CVD, which may miss some CVD-related deaths or mislabel patients who died from other heart diseases as CVD mortality. Mitigating its influence on our study motivated us to collect new data at MGH and evaluate our model using the surrogate gold standard CVD risk scores on the MGH dataset. Our results on the MGH dataset support the pre-trained model’s utilities on data from a different source.”

2) Comment #2:

A) For many clinicians, myself included, machine learning is a black box, and it is one that clinicians can never use. There is the stereotypical figure (Fig. 1A/1B) of data going in, “learned features” and then some sort of risk calculated (Fig. 1C). How does the machine actually do this? Is machine learning superior to other methods, e.g. logistic regression, random forest?

Response: In the revised manuscript, we have clarified the above points by adding more detailed descriptions on the workflow and redrawn Fig. 1 to illustrate the training and validation process clearly. The new content can be found from Page 2 Paragraph 4 to Page 4 Paragraph 1. For your convenience, the revised content is also attached below. In short, the deep learning model in our study was trained in an end-to-end fashion for CVD detection. The model extracts a high-dimensional feature vector from the input LDCT image. A distinctive difference between deep learning-based methods and classical machine learning methods like logistic regression and random forest lies in the feature definition and extraction. Classical methods require manual feature definition and extraction, which is also often referred to as feature engineering. The performance of classical machine learning methods is largely determined by the quality of hand-crafted features. In contrast, deep learning models automatically learn to extract relevant features in the training process. This data-driven feature extraction mechanism coupled with massive computing power and large-scale datasets makes deep learning outperform traditional machine learning methods in a large number of applications. In this study, we exploited the advantage of deep learning and the availability of large-scale clinical trial data to tackle the challenging problem of CVD risk stratification from LDCT scans.

To address the reviewer’s concern on model interpretability and transparency, we have made three additional major revisions. First, we have redesigned our model to explicitly incorporate an attention mechanism, which helps show the information being used. Second, we have included heatmaps demonstrating the focus areas of the network. The corresponding discussions have been updated in the revision. Third, besides releasing our source code, we packed the trained model into an open-access and ready-to-use tool so that others can test and verify our model (https://colab.research.google.com/github/DIAL-RPI/CVD-Risk-Estimator/blob/develop/colab_run.ipynb).

Revised Fig. 1 and the corresponding description on Page 2 Paragraph 4:

Figure 1: Overview of the proposed deep learning workflow. **a**, Public NLST and independent MGH datasets. The MGH dataset contains both LDCT and ECG-gated cardiac CT (CCT) images of 335 subjects. **b**, To extract an image volume of the heart, a deep CNN heart detector was developed and trained. **c**, A 3D CNN, Tri2D-Net, was designed for simultaneously screening CVD and predicting mortality risk. After training, the output probability of CVD positive also quantifies CVD mortality risk. **d**, The proposed model was further validated on the MGH dataset.

“Fig. 1 shows an overview of our study. Two datasets with a total of 10,730 subjects were included in our study (Fig. 1a). The public National Lung Screening Trial (NLST) dataset was used for model development and validation. It includes lung cancer screening LDCT exams of 10,395 subjects with abnormality records from the exam reports and causes of death for deceased subjects. An independent dataset collected at Massachusetts General Hospital (MGH) was used for further validation. Besides images and clinical reports of LDCT exams, the MGH dataset also collected ECG-gated cardiac CT of the same group of subjects, which enable us to calculate the clinically used CVD risk scores through clinical protocols for the validation. Our approach consists of two key components. First, a CNN heart detector was trained with 263 LDCTs from the NLST dataset to isolate the heart region (Fig. 1b). Second, we proposed a three-dimensional (3D) CNN model, Tri2D-Net, consisting of a CVD feature extractor and a CVD screening classifier was trained using CVD screening results as targeted labels (Fig. 1c). After training, the predicted probability of being CVD positive is used as a quantified CVD mortality risk score, which was validated by the CVD mortality labels on the NLST dataset. To further evaluate the generalization capability of our model, we calibrated the learned high-dimensional CVD features with three popular gold standard CVD risk scores, including, CAC score³³, CAD-RADS score³⁴, and MESA 10-year CHD risk score³⁵.”

Supplementary Fig. 1 for the improved model with attention mechanism and the corresponding description (Page 13 Paragraph 4):

Supplementary Fig. 1: **Framework of the proposed Tri2D-Net.** The network contains two stages: the feature extraction stage and the feature fusion stage. The feature extraction stage consists of three 2D CNN branches to extract features from the three sequences of orthogonal views. The feature fusion stage aggregates the extracted features for classification.

“As shown in Supplementary Fig. 1, the proposed Tri2D-Net consists of two stages, feature extraction and feature fusion. The former uses three 2D CNN branches to independently extract features from the three orthogonal views. By setting a cross-entropy loss for each branch, the three 2D CNN branches separately receive direct feedback to learn their parameters. This leads to a dramatically reduced optimization space compared to a massive 3D network⁴⁶. Specifically, in each branch, we split the original Resnet-18³⁵ into two parts, the first 13 layers (L13) and the last 5 layers (L-5). To introduce clinical prior knowledge into the network, we applied an attention block to help the model focus on calcification and fat regions, which highly correlate with CVD⁴⁸. The attention block first selects the calcification and fat regions with HU ranges (calcification: $HU > 130$; fat: HU in $[-190, -30]$). Then the masked slides are separately fed into a 4 layers CNN (first 4 layers of VGG11⁴⁹) to generate an attention map for each slide. The feature fusion module then concatenates the three feature representations extracted by the 2D CNN branches and feeds the result to a classifier for the final prediction.”

Heatmaps demonstrating the focus areas of the network (Fig. 5) and the corresponding case study on Page 10 Paragraph 3:

Figure 5: Visualizations of the features learned by the Tri2D-Net. The Gradient-weighted Class Activation Mapping (Grad-CAM³⁹), and the attention map learned by the attention block in the Tri2D-Net are visualized with heatmaps. The Grad-CAM³⁹ is a widely used visualization technique for CNN that produces a coarse localization map highlighting the important regions in the image for the final prediction.

“To interpret the prediction results of Tri2D-Net, we generated heatmaps using the Gradient-weighted Class Activation Mapping (Grad-CAM³⁹) and exported the attention maps from the attention block. Fig. 5 shows the results of three representative subjects from the NLST dataset. Fig. 5a&b belong to two subjects who died of CVD, referred as Case-a and Case-b, respectively. Fig. 5c shows the image of a subject Case-c, who survived by the end of the trial. Case-a has severe CAC with an average CAC Grade of 3.0 by the three radiologists. Tri2D-Net captured the strong calcium as shown in the Grad-CAM heatmap and predicted a high CVD risk score of 0.90. Case-b (Fig. 5b) had mild to moderate CAC with an average CAC Grade of 1.67. However, the attention block noticed abundant juxtacardiac fat and Tri2D-Net gave a high score of 0.82 for the case. Case-c has mild CAC graded as 1.33 by the radiologists. Since there was little calcification and juxtacardiac fat as indicated by the heatmaps, Tri2D-Net predicted a low risk score of 0.23 for this survived patient. Visualization of these cases demonstrates the contributions of both CAC and juxtacardiac fat to our model for predicting CVD risk scores in contrast with the mere reliance on CAC as the sole biomarker in prior studies²⁶⁻³⁰. The ability to capture various features for CVD makes our model superior to the existing CAC scoring models for CVD screening and CVD mortality quantification.”

B) Can the authors provide an example of a CT (like a video) where the machine performs this calculation? And could readers use this work? It would be great if a clinician like myself could pull up a CT of the chest through PACS, open it in the authors' program (or at least provide key slices for the program), and the program could provide a risk score of death from heart disease vs other causes. I think that for clinicians to use this in practice is the ultimate goal. With “standard” logistic regression, multivariate analysis, and nomogram, this could be estimated. Currently, as a reader of this work, it appears the authors have a good model, but it is unclear how it works, and I don't know how I can apply it to my own practice (though I would really like to).

Response: Thanks for the suggestion! We have released our source code on Github (<https://github.com/DIAL-RPI/CVD-Risk-Estimator>). Eight demos were included. The videos of four demos were added. To improve the usability, we have also updated the repository and packed the model into a ready-to-use tool powered by Google Colab (https://colab.research.google.com/github/DIAL-RPI/CVD-Risk-Estimator/blob/develop/colab_run.ipynb). The tool stores neither uploaded data nor patient information. The tool will estimate a stratified score from a LDCT scan showing both CVD probability and mortality risk.

● Minor Comments

1) *I would discuss more about the risk of death from competing causes in the discussion section. For example, the Nature Comms journal has published the following on this topic: PMID 32332714, 31729378*

Response: Thank you for your suggestion! We have discussed the heart disease-related fatalities in patients with cancer on Page 11 Paragraph 2 by referring to the two articles [41,42]:

“There are reports on increased heart disease-related fatalities of cancer patients in general, likely related to decreased cancer-specific mortality and aging population⁴¹. Older age-group, male gender, African American race, and unmarried status are key risk factors for cardiac mortality in cancer patients. Another study based on data from the Surveillance, Epidemiology, and End Results program reported higher risk of fatal stroke in patients with cancer⁴².”

2) *CVD is an overarching term that includes heart disease, stroke, aneurysm, etc. I recommend the authors just talk about heart disease.*

Response: Our previous analyses included all CVD related deaths including those related to heart disease, stroke, and other vascular causes due to the small number of death cases (within the 16,264 NLST subjects we had access to, there were only 1,521 death cases in total regardless of causes), limited access to the causes of death (only ICD-10 codes) and the details of relative abnormalities found on LDCT screening reports (only marked as “significant cardiovascular abnormality”). In our updated manuscript, we added a table showing the details of the selected CVD-related ICD-10 codes (Supplementary Table 1). What’s more, we also included an additional stratified analysis on patients with deaths related to heart disease only. The corresponding results are on Supplementary Fig. 3:

Heart Disease Mortality Prediction

	AUC	95% Confidence Interval	p Value
Our Model	0.757	0.734 - 0.801	-
KAMP-Net	0.684	0.628 - 0.714	0.0025
AE+SVM	0.650	0.605 - 0.695	0.0001

Supplementary Fig. 3: **Experimental results on the NLST dataset for heart disease only mortality prediction.** In stead of using all 17 ICD-10 codes in Supplementary Table 1, only heart disease related (I11, I20, I21, I24, and I25) deceases are regarded as positive cases (86 subjects). Our model significantly exceeds other methods ($P \leq 0.0025$) and achieved a performance similar to the average performance of human experts.

Supplementary Table 1: Selected CVD-related causes of death.

Index	ICD-10 Code	Detail
1	I10	Essential (primary) hypertension
2	I11	Hypertensive heart disease
3	I12	Hypertensive renal disease
4	I13	Hypertensive heart and renal disease
5	I20	Angina pectoris
6	I21	Acute myocardial infarction
7	I24	Other acute ischemic heart diseases
8	I25	Chronic ischemic heart disease
9	I63	Cerebral infarction
10	I64	Stroke, not specified as haemorrhage or infarction
11	I65	Occlusion and stenosis of precerebral arteries
12	I66	Occlusion and stenosis of cerebral arteries
13	I67	Other cerebrovascular diseases
14	I70	Atherosclerosis
15	I71	Aortic aneurysm and dissection
16	I72	Other aneurysm
17	I73	Other peripheral vascular diseases

3) *Could the authors please discuss how these CTs could be used to follow pediatric cancer patients to predict competing death? For example, PMID 32298481*

Response: The approach can be extended to other CT exams if sufficient data can be collected and labeled, such as for pediatric cancer patients in [40]. We have mentioned such a possibility on Page 11 Paragraph 2:

“It is worth noting that the proposed method demonstrated on this specific application may apply to other clinical studies, for example, predicting the cause of death among pediatric patients with cancer⁴⁰.”

4) *The grammar/syntax needs to be corrected in a few areas, eg, “and achieved the state-of-the-art performance”*

Response: Thanks for pointing out the problems! We have checked the paper and fixed these problems.

5) *Also, state of art performance is very subjective. I would delete this.*

Response: We have deleted such expressions in our revision.

6) *There are references in the abstract, and this cannot be present.*

Response: Done.

Response to Reviewer#2:

In this paper, the authors present a deep learning model for CVD risk analysis from LDCT collected in a lung cancer screening setting. The deep learning method extracts the heart area from the CT scan, and performs an analysis of that area to produce a binary score whether signs of CVD are present or not. A subject was considered CVD-positive if any cardiovascular abnormality was reported in the subject's CT screening exams or the subject died of CVD. A CVD-negative subject has no CVD related medical history and no reported cardiovascular abnormality in any of the CT scans during the trial, and did not die of circulatory system diseases. The authors trained their system using data from the NLST trial, for which they also set 20%, a set of 2085 subjects, aside for testing. Finally, a retrospective dataset of 119 subjects is collected from MGH and the performance of the model is evaluated against the three clinical standards for measuring CVD from cardiac CT: CAC scores, CAD-RADS scores, and MESA scores.

● Major Comments

- 1) The paper is unclear about what the deep learning model precisely predicts. Based on my review of the paper, I think the model is primarily intended to predict whether there are signs of CVD or not. However, this is quite vague in the abstract and needs to be clarified. This model is not directly trained for mortality prediction, I think. However, page 3 reads "A deep CNN model for CVD mortality risk analysis of LDCT images was designed and trained on the NLST dataset". Further on in this paragraph, it reads that the model was trained for CVD positive or CVD negative, so this is unclear and really needs clarification. If the model intends to predict mortality, it needs a time frame needs to be attached to that. Mortality within five years, one year? 10 years?*

Response: Sorry for the confusion. The deep learning model in our study was first trained for CVD screening. The output of the deep learning model is a probability in the range of 0 to 1, with 0 being very unlikely and 1 being very likely for a patient to have CVD. We then considered this same output value as a risk score for mortality prediction. In our study, we evaluated the potential of this risk score by predicting the mortality of patients within 6 years, which is the length of the follow-up period of the NLST. In the revision, we have clarified these two deliverables on Page 2, Paragraph 3:

*"To tackle the limitations of the prior studies, we built an end-to-end deep neural network to **a)** screen patients for CVDs and **b)** quantify CVD mortality risk scores directly from chest LDCT examinations. Specifically, our approach focuses on the cardiac region in a chest LDCT scan and makes predictions based on automatically learned comprehensive features of CVDs and mortality risks. The prediction is calibrated against the incidence of CVD abnormalities during the follow-up period of a clinical trial, subjective assessment of radiologists in reader studies, and the CVD risk scores calculated from electrocardiogram (ECG)-gated cardiac CT including the CAC score³³, CAD-RADS score³⁴, and MESA 10-year CHD risk score³⁵."*

To illustrate the training and validation workflow of our work more clearly, we also redrew the Fig. 1 and added more detailed description from Page 2 to Page 3:

Figure 1: **Overview of the proposed deep learning workflow.** **a**, Public NLST and independent MGH datasets. The MGH dataset contains both LDCT and ECG-gated cardiac CT (CCT) images of 335 subjects. **b**, To extract an image volume of the heart, a deep CNN heart detector was developed and trained. **c**, A 3D CNN, Tri2D-Net, was designed for simultaneously screening CVD and predicting mortality risk. After training, the output probability of CVD positive also quantifies CVD mortality risk. **d**, The proposed model was further validated on the MGH dataset.

“Fig. 1 shows an overview of our study. Two datasets with a total of 10,730 subjects were included in our study (Fig. 1a). The public National Lung Screening Trial (NLST) dataset was used for model development and validation. It includes lung cancer screening LDCT exams of 10,395 subjects with abnormality records from the exam reports and causes of death for deceased subjects. An independent dataset collected at Massachusetts General Hospital (MGH) was used for further validation. Besides images and clinical reports of LDCT exams, the MGH dataset also collected ECG-gated cardiac CT of the same group of subjects, which enable us to calculate the clinically used CVD risk scores through clinical protocols for the validation. Our approach consists of two key components. First, a CNN heart detector was trained with 263 LDCTs from the NLST dataset to isolate the heart region (Fig. 1b). Second, we proposed a three-dimensional (3D) CNN model, Tri2D-Net, consisting of a CVD feature extractor and a CVD screening classifier was trained using CVD screening results as targeted labels (Fig. 1c). After training, the predicted probability of being CVD positive is used as a quantified CVD mortality risk score, which was validated by the CVD mortality labels on the NLST dataset. To further evaluate the generalization capability of our model, we calibrated the learned high-dimensional CVD features with three popular gold standard CVD risk scores, including, CAC score³³, CAD-RADS score³⁴, and MESA 10-year CHD risk score³⁵.”

2) Comments#2:

A) *The external validation is limited - a relatively small dataset of 119 patients is retrospectively collected.*

Response: We have made our best efforts to expand the external validation dataset from 106 to 335 patients, including 181 CVD positive and 154 CVD negative patients. On this expanded validation dataset, we have conducted a new experiment of CVD screening by directly applying the model trained on the NLST dataset without any tuning. The updated results from this experiment are shown in Fig. 4a (shown in the response to the part B) of this comment). The detailed analysis is on Page 7 Paragraph 3:

“In the experiment of CVD screening shown in Fig. 4a, our deep learning model achieved a significantly higher ($p < 0.0001$) AUC value of 0.924 (95% confidence interval, 0.909-0.940) than its performance on the NLST dataset (0.871), where the network was originally trained. Superior performance on this external MGH dataset may be due to the better image quality of the contemporary MGH dataset acquired from newer scanners. Furthermore, the MGH annotated datasets and use of ECG-gated cardiac CT as a gold standard may have been superior to the annotation in NLST. This experiment shows that the proposed deep learning model has good generalizability and is highly consistent with human experts' joint label for CVD screening from LDCT and CVD scores calculated on ECG-gated cardiac CT.”

The description of the this expanded MGH dataset is as follows (Page 4 Paragraph 4):

“Furthermore, through an institutional review board (IRB) approved retrospective study, we acquired an independent and fully de-identified dataset from MGH in 2020. This MGH dataset contains 335 patients (161 men, 174 women, mean age 63.6 ± 8.0 years), who underwent LDCT for lung cancer screening. Of these, 100 patients had no observed CVD abnormalities in their LDCT. The remaining 235 subjects underwent ECG-gated cardiac CT for CVD risk assessment for atypical chest pain, equivocal stress test, and chest pain with low to intermediate risk for CHD. Three CVD risk scores were calculated for the 235 subjects from their cardiac CT images, including CAC score³³, coronary stenosis (quantified as CAD-RADS)³⁴, and MESA 10-year CHD risk³⁵. Table 1 lists the characteristics of the dataset (see Methods, MGH dataset). The MGH dataset was used to evaluate the clinical significance of the NLST-trained model for feature extraction without re-training or fine-tuning. For the validation of CVD screening, a subject was labeled as CVD-positive if the subject underwent an ECG-gated cardiac CT screening and received either a CAC score > 10 or a CAD-RADS score > 1 . Correspondingly, a CVD-negative subject had either all the LDCT screening exams being negative or the calculated scores of $CAC \leq 10$ and $CAD-RADS \leq 1$. Based on the above criteria, 181 subjects were CVD positive and 154 patients were CVD negative. For the quantification of CVD mortality risk, since this MGH dataset was collected from a patient cohort, who had recent exams (2015-2020), the mortality records are not yet available. We instead calibrate our model against the three gold standard risk scores as surrogate evaluators.”

B) *In addition, the authors retrained a logistic regression model using 5-fold CV using the features from the deep learning model? Why? In this way, it is not a direct external validation of the model developed on the NLST data, but more a validation of the feature extractor part of the DL model. Also, it is not completely clear what features are used. The output of the fully connected layer? How many features are there in this feature representation? Since a five-fold CV is performed, are the selected features for the logistic regression stable?*

Response: We thank the reviewer for pointing out the problem. In the revised article, we have added a new CVD screening experiment by directly applying the trained model from NLST to the expanded MGH dataset as a direct external validation. We have also updated the previous experiments' results, which aim to validate whether the trained CVD feature extractor can effectively extract features highly related to CVD mortality risks. Since we do not have the mortality records, we have calibrated the deep features extracted from LDCT against the gold standard CVD scores calculated with the established clinical procedures based on the ECG-gated coronary angiograms. In the revision, we have provided all details about this experiment. The features used in all three experiments are from the last fully-connected layer, which is a 1,536-dimensional feature vector. All these 1,536 dimensions are directly used by the logistic regression models in the 5-fold CV experiments. That is, the feature selection in three experiments are consistent. We have included the key points and updated results in the revision (Page 9 to Page 10):

“To evaluate the generalization ability of the deep learning quantified CVD risk score, we directly applied the trained CVD feature extractor to the MGH data and used the extracted high-dimensional feature to estimate the three gold standard risk scores calculated from ECG-gated cardiac CT for comparison through linear classifiers (logistic regression). Five-fold cross-validation was used in all three experiments to fit and validate the three linear classifiers. Note that in all three experiments, the CVD feature extractor parameters are fixed, and the linear classifiers are directly applied to the entire high-dimensional feature without dimension reduction or feature selection. It is a standard procedure to evaluate whether the feature extractor can effectively extract features highly relative to downstream tasks³⁶⁻³⁸. Our model was compared with the radiologists' annotated CAC grades and the two other previously reported studies on CVD risk prediction^{31,32}. Like in the NLST experiments, the CAC grades performance under the settings of 1+, 2+ and 3 was calculated. Our experimental results are presented as follows.

The features extracted by the trained CVD feature extractor from LDCT were first used to estimate the CAC score³³. With a threshold of 400 for CAC scores, the MGH subjects were divided into two groups: 78 subjects with severe CAC and 157 subjects with none or minor CAC. Our model achieved an AUC value of 0.942 (95% confidence interval, 0.927-0.958) and significantly outperformed the other two methods ($p < 0.0001$, see Fig. 4b). It is worth noting that our model is competitive with experienced radiologists. The AUC of our DL model was only slightly lower than the radiologists' CAC grading without significant difference ($p > 0.43$), despite the fact that our DL model has never been trained for CAC score estimation. These results suggest that deep learning analysis of LDCT can well approximate the human expert performance using CCT in differentiating patients with severe and non-severe CAC.

The second experiment evaluates the capability of the deep learning model in classifying subjects into high and low risk groups using LDCT by comparing against the coronary stenosis (CAD-RADS) scores³⁴ obtained by human experts on CCT. Subjects with CAD-RADS scores greater than or equal to 4 are labeled as with severe stenosis, i.e., positive samples (51 subjects). The other 184 subjects with smaller scores were labeled as negative. Our model reached an AUC value of 0.808 (95% confidence interval, 0.758-0.858, see Fig. 4c). Our model significantly outperformed the other two methods ($p \leq 0.0338$). Unlike calcification, coronary stenosis is much harder to detect through a chest LDCT screening, while it is a direct biomarker of CVD risk. The performance obtained using LDCT is thus highly encouraging. The superiority demonstrates that our model can quantify the subclinical imaging markers on LDCT, making it a promising tool for CVD assessment in lung cancer screening.

Figure 4: **Results of 4 experiments on the MGH dataset.** **a**, Validation of CVD screening. **b & c & d**, Comparison of our deep learning model with three clinical standard criteria the CAC score³³ (**b**), CAD-RADS³⁴ (**c**), and MESA 10-year risk score³⁵ (**d**) calculated with the standard protocol. For radiologists' CAC grades in **b**, red dots represents CAC Grade 1+, yellow dots represents CAC Grade 2+, green dots represents CAC Grade 3.

In the third experiment, patients were divided into high and low risk groups according to MESA 10-year risk score³⁵, which is a clinical gold-standard risk stratification score for CVD integrating multiple factors including gender, age, race, smoking habit, family history, diabetes, lipid lowering and hypertension medication, CAC score extracted from CCT, and laboratory findings including cholesterol and blood pressure. Because some of the 235 subjects did not have all the needed exams, we are only able to calculate the MESA scores of 106 subjects. When median MESA 10-year risk score in our patients was used as a threshold (14.2), 52 subjects with greater scores were labeled as high risk, while the other 54 subjects were labeled as low risk. Our model achieved an AUC value of 0.799 (95% confidence interval, 0.736-0.863), which significantly outperformed all the other methods (see Fig. 4d)."

3) *For the NLST cases, what CT scan is used? This is unclear. If all CT volumes are used, how do the authors get one score per patient? Is the maximum deep learning score across the different volumes used? Why not perform an analysis per CT when assessing the performance of the model to detect signs of CVD?*

Response: Thanks for the great question. In the training phase, all candidate CT volumes were treated as independent cases to increase training samples. In the validation and testing phases, a single CT volume was randomly selected from the candidate volumes of each subject to preserve the original data distribution. Since the number of CT scans is not consistent across the subjects, the use of these CT scans as independent cases would change the data distribution and introduce a bias to the test results. We have not included any ensemble methods to merge predictions on multiple cases of the same subject since the NLST dataset was not prepared for such a study. The situation of CVD positive subjects is especially complex. For CT scans of one subject, there are cases for which only the second scan was labeled as abnormal, but no CVD abnormalities were reported on the first and third scans. Hence, we have only focused on the normal and abnormal scans with clear labels. The changes over time were not considered in this work. In the revision, we have further clarified these issues on Page 13 Paragraph 4 and more detailed inclusion and exclusion criteria in the case of the NLST dataset can be found on Supplementary Fig. 2.

“Since the NLST was designed for lung cancer screening, CVD related information is incomplete. Therefore, we used LDCT from only those patients with clear CVD information. In the training set, all LDCT volumes generated in the candidate exams were treated as independent cases to increase the number of training samples. In the validation and testing phases, a single CT volume was randomly selected from the candidate volumes of each subject to preserve the original data distribution. Since the number of LDCT exams is inconsistent across subjects (for example, patients with death or diagnosis of lung cancer on initial LDCTs did not complete all follow-up LDCTs), the use of these CT volumes as independent cases can change the data distribution and introduce a bias to the test results.”

Supplementary Fig. 2: STARD flow diagram of the inclusion and exclusion of images in the NLST dataset used in our analysis.

4) *The abstract and introduction are too strong with regards to the benefits of using LDCT for CVD screening. At present, there is no scientific consensus whether the benefits of LDCT screening for CVD outweigh the harms. LDCT screening has only shown benefit for reducing lung cancer mortality in a specific high-risk population. The introduction and abstract should be changed to reflect this and explain the current situation better.*

Response: Both the abstract and introduction have been revised. The new abstract reads as follows:

“Cancer patients have a higher risk of cardiovascular disease (CVD) mortality than the general population. Low dose computed tomography (LDCT) for lung cancer screening offers

an opportunity for simultaneous CVD risk estimation in at-risk patients. Our deep learning CVD risk prediction model, trained with 30,286 LDCTs from the National Lung Cancer Screening Trial, achieved an area under the curve (AUC) of 0.871 on a separate test set of 2,085 subjects and identified patients with high CVD mortality risks (AUC of 0.768). We validated our model against ECG-gated cardiac CT based markers including coronary artery calcification (CAC) score, CAD-RADS score, and MESA 10-year CHD risk score from an independent dataset of 335 subjects. Our work shows that, in high-risk patients, deep learning can convert LDCT for lung cancer screening into a dual-screening quantitative tool for CVD risk estimation.”

● **Minor Comments**

1) *The NLST is quite old data. What effect does this have on the model?*

Response: Thank you for your comment. Compared to a somewhat historic data from NLST, we had more contemporary data from MGH from 2015-2020 acquired on modern MDCT scanners including GE 750 HD, GE Revolution, Siemens Definition Edge, Force, and Flash scanners. LDCT images in the NLST dataset contain stronger noise than those recently collected at MGH. However, training on such noisy data helps produce a robust and more generalizable model. Given the improved accuracy, we do not think that the old NLST adversely affected the performance of our algorithm on the new data from MGH. The corresponding analysis can be found in the response to the major comment #1).

2) *The claims in the Discussion need to be toned down. No prospective data yet, so this is only preliminary evidence.*

Response: We have revised the Discussions to reflect the point (Page 10 Paragraph 2). Thanks!

“In summary, our deep learning model demonstrates the value of lung cancer screening LDCT for CVD risk estimation. Given the increasing utilization of LDCT-based lung cancer screening, shared risk factors, and high prevalence of CVD in these at-risk patients, the potential of obtaining a quantitative and reliable CVD risk score by analyzing the same scans may benefit a large patient population. The further comparative study on the deep learning model of LDCT images with human experts on CCT for risk group classification shows that the deep learning model can analyze LDCTs to achieve performance approximating the clinical reading with dedicated cardiac CTs. The comparable or superior performance of our model from LDCT implies that additional dedicated ECG-gated coronary calcium scoring and other laboratory tests could be avoidable. Our deep learning model may thus help reduce the cost and radiation dose in the workup of at-risk patients for CVD with quantitative information from a single LDCT exam. Given the technical challenges associated with the quantification of CAC from LDCT for lung cancer screening versus ECG-gated CCT, our study indicates a significant development in establishing a CVD-related risk nomogram with LDCT.”

3) *The deep learning model mostly focuses on the CAC, it seems, as the AUC is 0.942 for this, while lower for MESA and CAD-RADS (0.817 and 0.809 respectively). Isn't this method primarily picking up the calcifications in the heart? Any analysis done to check this?*

Response: Since calcium is the most salient and relevant feature to CVD, it would not be a surprise that our model pays significant attention to calcium in the heart region. To study whether our model utilized features other than calcium, we have performed case studies in the revised paper. Based on the visualization of the learned features, our model can also capture

epicardial fat, which is also an important marker of underlying coronary artery disease⁴⁸. The details of the case studies are on Page 10 Paragraph 3:

Figure 5: Visualizations of the features learned by the Tri2D-Net. The Gradient-weighted Class Activation Mapping (Grad-CAM³⁹), and the attention map learned by the attention block in the Tri2D-Net are visualized with heatmaps. The Grad-CAM³⁹ is a widely used visualization technique for CNN that produces a coarse localization map highlighting the important regions in the image for the final prediction.

“To interpret the prediction results of Tri2D-Net, we generated heatmaps using the Gradient-weighted Class Activation Mapping (Grad-CAM³⁹) and exported the attention maps from the attention block. Fig. 5 shows the results of three representative subjects from the NLST dataset. Fig. 5a&b belong to two subjects who died of CVD, referred as Case-a and Case-b, respectively. Fig. 5c shows the image of a subject Case-c, who survived by the end of the trial. Case-a has severe CAC with an average CAC Grade of 3.0 by the three radiologists. Tri2D-Net captured the strong calcium as shown in the Grad-CAM heatmap and predicted a high CVD risk score of 0.90. Case-b (Fig. 5b) had mild to moderate CAC with an average CAC Grade of 1.67. However, the attention block noticed abundant juxtacardiac fat and Tri2D-Net gave a high score of 0.82 for the case. Case-c has mild CAC graded as 1.33 by the radiologists. Since there was little calcification and juxtacardiac fat as indicated by the heatmaps, Tri2D-Net predicted a low risk score of 0.23 for this survived patient. Visualization of these cases demonstrates the contributions of both CAC and juxtacardiac fat to our model for predicting CVD risk scores in contrast with the mere reliance on CAC as the sole biomarker in prior studies²⁶⁻³⁰. The ability to capture various features for CVD makes our model superior to the existing CAC scoring models for CVD screening and CVD mortality quantification.”

4) Since reader group 2, consisting of only one reader, performed better, please split out the experience levels of the three readers (M.K.K., R.S. and R.D.K.). Is there a difference in experience level?

Response: For a more convincing comparison, M.K.K and R.S. have independently reviewed the whole test set, leading to three sets of independent reviews. The updated results are now summarized in Fig. 2, along with the corresponding analysis from Page 5 to Page 7. Furthermore, we included Kaplan Meier curves to compare the performance of our model to the radiologists.:

“Two experiments were conducted on the NLST dataset for the evaluation of CVD screening and CVD mortality quantification, respectively, where the proposed deep learning model was compared with other deep learning models and against CAC grades read by radiologists. Three radiologists from MGH (M.K.K., R.S. and R.D.K.) with 2-15 years of clinical experience averaged at 7 years, independently graded all the 2,085 CT volumes to obtain the CAC grades. Four CAC categories were used, including no calcification (level 0 - normal), calcification over less than 1/3 of the length of coronary arteries (level 1 - minimal), calcification over 1/3 to 2/3 of the coronary arterial lengths (level 2 - moderate) and calcification greater than 2/3 of the arterial length (level 3 - heavy). The average reader results are calculated by averaging the results of the three radiologists.

Figure 2: **Experimental results on the NLST dataset.** **a**, Reader study and comparison of our model with other reported methods on CVD detection. **b**, Reader study and comparison of our model with other reported methods on CVD caused mortality prediction. For reader study, green symbols represents CAC Grade 1+, yellow symbols represents CAC Grade 2+, red symbols represents CAC Grade 3.

We first evaluated the proposed model for identifying patients with CVDs from the lung cancer screening population. Fig. 2a shows the receiver operating characteristic curves (ROCs) of multiple methods. Our deep learning model achieved an area under the curve (AUC) of 0.871 (95% confidence interval, 0.860-0.882)(see Methods, Statistical analysis). With a positive predictive value (PPV) of 50.00%, the model achieved a sensitivity of 87.69%, which suggests that our model can identify 87.69% of the CVD-positive subjects using only a chest LDCT scan, when allowing half of the positive predictions as false. For the reader performance, all patients with \geq minimal CAC (CAC Grade 1+) are considered as abnormal. It can be seen in Fig. 2a that CAC Grade 1+ yielded a sensitivity of 96.6% and a PPV of 35.3%. With a similar sensitivity of 96.6%, our model achieved a slightly but not significantly higher PPV of 38.4% ($p=0.3847$). In addition, we compared our model with the two recently reported works, KAMP-Net³¹ and Auto-encoder (AE+SVM)³⁰. The table in Fig. 2a shows that our model significantly outperformed the other two methods ($p<0.0001$). It indicates that our model can differentiate subjects with high CVD risk from those with low risk from LDCT.

Figure 3: **Kaplan Meier curves on the NLST dataset.** Comparison of our model with the radiologists. Thresholds were selected on the CVD risk score calculated by our model to enforce the low (a) or high (b) risk group has a similar survival probability with that of the radiologists.

Further, we evaluated the performance of our model in quantification of CVD mortality risk. The results are shown in Fig. 2b, where CAC Grades 1+, 2+, and 3 denote the performance of mortality prediction using extent of subjective CAC categories 1 and above, categories 2 and above, and 3 only, respectively. CAC Grades 1+, 2+, and 3 denote the performance of mortality prediction using categories 1 and above, categories 2 and above, and 3 only, respectively. The trained deep learning model was directly applied to this testing set to predict the CVD-caused mortality without fine-tuning. Our deep learning model achieved an AUC value of 0.768 (95% confidence interval, 0.734-0.801), which significantly outperformed the competing methods ($p < 0.0001$) as shown in Fig. 2b. With the same PPV of averaged CAC Grade 2+ (10.8%), our model achieved a sensitivity of 80.8%. Specifically, in the NLST test set, our model successfully identified 97 of the 120 deceased subjects as high risk, while the averaged CAC Grade 2+ labeled 35 of those 97 cases as low risk. Additionally, it can be seen from Fig. 2b that our model achieved a similar performance to the average performance of human experts. Although the performance of reader 1 is higher than our model, there is a significant difference between the two groups' annotation ($p < 0.0001$). For further comparison, Fig. 3 shows the Kaplan Meier curves of different risk groups labeled by our model and the radiologists, respectively. For the radiologists, we used the average reader prediction of CAC Grade 2+ to separate the subjects into low/high risk groups and drew the Kaplan Meier curves for both groups. The final survival probabilities of low and high risk groups by radiologists are 95.79% and 85.83%, respectively. For fair and direct comparison, in Fig. 3a, we selected a threshold to divide the quantified CVD risk scores using our model so that the low risk group has a survival probability of 95.79%, similar to the radiologists. Under this circumstance, the model-predicted high-risk group showed a significantly lower ($p=0.0059$) final survival probability of 73.24%. Similarly, in Fig. 3b, we selected a threshold so that the high-risk group has a survival probability of 85.75%, also similar to the radiologists. In this case, the model-predicted low-risk group achieved a significantly higher ($p=0.0272$) final survival probability of 97.46%. Thus, our model can help reduce inter-and intra-observer variations in quantifying CAC. The model can also automatically categorize CVD risks so that radiologists can focus on other tasks such as lung nodule detection, measurement, stability assessment, classification (based on nodule attenuation), and other incidental findings in the chest and upper abdomen."

5) *The authors write "Then, a max pooling is employed on the slide dimension to fuse all these feature maps into one feature map". So, Resnet-18 has 256 feature maps after L13. The feature map size is 14x14x256xz, where z is the number of slices. Then, a max pool is done over the z-direction, which leads to a new feature map of 14x14x256? If yes, this sentence should be rephrased because it reads 'one feature map', which is confusing.*

Response: Yes, the understanding is correct. We have rephrased this sentence and moved details of the model to the supplementary information Page S1 Paragraph 1:

"Then a max pooling operation is employed along the dimension of slices to merge the feature maps of all slices into a single new feature map."

6) *How were the heatmaps generated? This needs to be clarified.*

Response: The heatmaps were generated using Grad-CAM³⁹, a widely used visualization technique for CNN that produces a coarse localization map highlighting the important regions in the image for the final prediction. We provide the details in the response to comment #3) above.

Response to Reviewer#3:

● Major Comments

- 1) *The NLST dataset were labeled as normal or abnormal based on LDCT report, medical history, and cause of mortality. Since cardiovascular abnormalities on LDCT or cardiovascular diseases have wide spectrum of severity or clinical relevance (e.g. from tiny coronary calcification to overt sign of heart failure, and from mild arterial hypertension to myocardial infarction), the authors need to provide clear definition of abnormal label, or actual spectrum of LDCT abnormalities or cardiovascular diseases in the NLST dataset.*

Response: For the LDCT reports, we looked for the note of “significant cardiovascular abnormalities” recorded in the dataset. If a subject has such a note on any of the LDCT exams, the subject is labeled as CVD positive. For the cause of mortality, based on the ICD-10 codes recorded in the dataset, we selected 17 codes related to CVD. The 17 codes are listed in Supplementary Table 1. All the deceased subjects with a CVD-related ICD-10 code are labeled as CVD positive. For the medical history, we used the histories of heart disease and hypertension reported in the dataset. A subject is labeled as CVD negative only if 1) the subject had no abnormalities reported in any of the LDCT exams, 2) did not die during the trial because of CVD and 3) did not have any known history of CVD. We excluded subjects with history of heart disease, heart attack, stroke and hypertension from the CVD negative list.

We agree with the reviewer that NLST labels can be misleading and difficult to categorize. To overcome the labelling issues, as stated in the revised manuscript, three radiologists separately graded CAC on 2,085 LDCT CTs from NLST. Our model performed favorably compared to these radiologists for both CVD screening and CVD mortality risk estimation. A detailed description of CAC grading is given on Page 6 Paragraph 1, which is also attached below. More importantly, the independent MGH dataset was much better labeled and includes three CVD risk scores were calculated from the subjects’ cardiac CT images, including CAC score, coronary stenosis (quantified as CAD-RADS), and MESA 10-year CHD risk. With this external validation MGH dataset, the effectiveness of our model has been clearly demonstrated by its strong performance in all the four experiments on these well-controlled and clearly-labeled external data.

Supplementary Table 1: Selected CVD-related causes of death.

Index	ICD-10 Code	Detail
1	I10	Essential (primary) hypertension
2	I11	Hypertensive heart disease
3	I12	Hypertensive renal disease
4	I13	Hypertensive heart and renal disease
5	I20	Angina pectoris
6	I21	Acute myocardial infarction
7	I24	Other acute ischemic heart diseases
8	I25	Chronic ischemic heart disease
9	I63	Cerebral infarction
10	I64	Stroke, not specified as haemorrhage or infarction
11	I65	Occlusion and stenosis of precerebral arteries
12	I66	Occlusion and stenosis of cerebral arteries
13	I67	Other cerebrovascular diseases
14	I70	Atherosclerosis
15	I71	Aortic aneurysm and dissection
16	I72	Other aneurysm
17	I73	Other peripheral vascular diseases

2) Regarding the interpretation by radiologists, radiologists focused on the presence and extent of coronary artery calcifications, for prediction of cardiovascular disease or related mortality. However, there may other abnormalities suggesting cardiovascular diseases such as enlarged heart chamber, dilatation or calcification of aorta, and pleural or pericardial effusion, and the performance of radiologists for identification of cardiovascular diseases may have been underestimated

Response: We agree with the reviewer that other cardiac findings as stated in the comment can also predict CVD related risks. Although our network may implicitly learn these features in the data-driven manner, we plan to train our model to look for other findings as you stated in your comment (such as pericardial effusion, pleural effusion, cardiac chamber dilation, aortic dimensions). Our review revealed that there were very few cases with these findings (<1%) in the MGH dataset to make a statistically meaningful evaluation. With an enriched dataset, in future, we intend to have radiologists assess these findings as well.

Based on the cases we have now and aided by visualization, we have shown that our model did use features other than calcification like excessive of fat to help estimate CVD risks. The corresponding analysis is shown in the response to the next question.

3) The authors may provide representative cases along with the heat maps from the algorithm, to demonstrate whether the algorithm actually focused on findings of cardiovascular diseases (e.g. coronary calcification).

Response: Thanks for the suggestion! We have added Fig. 5 to visualize representative cases, along with the corresponding analysis (Page 10 Paragraph 3):-

Figure 5: **Visualizations of the features learned by the Tri2D-Net.** The Gradient-weighted Class Activation Mapping (Grad-CAM³⁹), and the attention map learned by the attention block in the Tri2D-Net are visualized with heatmaps. The Grad-CAM³⁹ is a widely used visualization technique for CNN that produces a coarse localization map highlighting the important regions in the image for the final prediction.

“To interpret the prediction results of Tri2D-Net, we generated heatmaps using the Gradient-weighted Class Activation Mapping (Grad-CAM³⁹) and exported the attention maps from the attention block. Fig. 5 shows the results of three representative subjects from the NLST dataset. Fig. 5a&b belong to two subjects who died of CVD, referred as Case-a and Case-b, respectively. Fig. 5c shows the image of a subject Case-c, who survived by the end of the trial.

Case-a has severe CAC with an average CAC Grade of 3.0 by the three radiologists. Tri2D-Net captured the strong calcium as shown in the Grad-CAM heatmap and predicted a high CVD risk score of 0.90. Case-b (Fig. 5b) had mild to moderate CAC with an average CAC Grade of 1.67. However, the attention block noticed abundant juxtacardiac fat and Tri2D-Net gave a high score of 0.82 for the case. Case-c has mild CAC graded as 1.33 by the radiologists. Since there was little calcification and juxtacardiac fat as indicated by the heatmaps, Tri2D-Net predicted a low risk score of 0.23 for this survived patient. Visualization of these cases demonstrates the contributions of both CAC and juxtacardiac fat to our model for predicting CVD risk scores in contrast with the mere reliance on CAC as the sole biomarker in prior studies²⁶⁻³⁰. The ability to capture various features for CVD makes our model superior to the existing CAC scoring models for CVD screening and CVD mortality quantification.”

4) *For cardiovascular mortality prediction in the NLST dataset, performance of radiologists exhibited substantial difference. I wonder whether there was difference in the experience of radiologists and less-experienced radiologist exhibited lower performance.*

Response: For a more convincing comparison, M.K.K and R.S. have independently reviewed the whole test set, leading to three sets of independent reviews. The three readers are all from MGH and have 2-15 years of clinical experience averaged at 7 years. We have included this information on Page 5 Paragraph 2:

“Three radiologists from MGH (M.K.K., R.S. and R.D.K.) with 2 to 15 years of clinical experience averaged at 7 years, independently graded all the 2,085 CT volumes to obtain the CAC grades.”

The updated results are now shown in Fig. 2, along with the corresponding analysis from Page 5 to page 7. What’s more, we also included Kaplan Meier curves to further compare our model with the radiologists:

“We first evaluated the proposed model for identifying patients with CVDs from the lung cancer screening population. Fig. 2a shows the receiver operating characteristic curves (ROCs) of multiple methods. Our deep learning model achieved an area under the curve (AUC) of 0.871 (95% confidence interval, 0.860-0.882)(see Methods, Statistical analysis). With a positive predictive value (PPV) of 50.00%, the model achieved a sensitivity of 87.69%, which suggests that our model can identify 87.69% of the CVD-positive subjects using only a chest LDCT scan, when allowing half of the positive predictions as false. For the reader performance, all patients with \geq minimal CAC (CAC Grade 1+) are considered as abnormal. It can be seen in Fig. 2a that CAC Grade 1+ yielded a sensitivity of 96.6% and a PPV of 35.3%. With a similar sensitivity of 96.6%, our model achieved a slightly but not significantly higher PPV of 38.4% ($p=0.3847$). In addition, we compared our model with the two recently reported works, KAMP-Net³¹ and Auto-encoder (AE+SVM)³⁰. The table in Fig. 2a shows that our model significantly outperformed the other two methods ($p<0.0001$). It indicates that our model can differentiate subjects with high CVD risk from those with low risk from LDCT.

Figure 2: **Experimental results on the NLST dataset.** **a**, Reader study and comparison of our model with other reported methods on CVD detection. **b**, Reader study and comparison of our model with other reported methods on CVD caused mortality prediction. For reader study, green symbols represents CAC Grade 1+, yellow symbols represents CAC Grade 2+, red symbols represents CAC Grade 3.

Figure 3: **Kaplan Meier curves on the NLST dataset.** Comparison of our model with the radiologists. Thresholds were selected on the CVD risk score calculated by our model to enforce the low **(a)** or high **(b)** risk group has a similar survival probability with that of the radiologists.

Further, we evaluated the performance of our model in quantification of CVD mortality risk. The results are shown in Fig. 2b, where CAC Grades 1+, 2+, and 3 denote the performance of mortality prediction using extent of subjective CAC categories 1 and above, categories 2 and above, and 3 only, respectively. CAC Grades 1+, 2+, and 3 denote the performance of mortality prediction using categories 1 and above, categories 2 and above, and 3 only, respectively. The trained deep learning model was directly applied to this testing set to predict the CVD-caused mortality without fine-tuning. Our deep learning model achieved an AUC value of 0.768 (95% confidence interval, 0.734-0.801), which significantly outperformed the competing methods ($p < 0.0001$) as shown in Fig. 2b. With the same PPV of averaged CAC Grade 2+ (10.8%), our model achieved a sensitivity of 80.8%. Specifically, in the NLST test set, our model successfully identified 97 of the 120 deceased subjects as high risk, while the

averaged CAC Grade 2+ labeled 35 of those 97 cases as low risk. Additionally, it can be seen from Fig. 2b that our model achieved a similar performance to the average performance of human experts. Although the performance of reader 1 is higher than our model, there is a significant difference between the two groups' annotation ($p < 0.0001$). For further comparison, Fig. 3 shows the Kaplan Meier curves of different risk groups labeled by our model and the radiologists, respectively. For the radiologists, we used the average reader prediction of CAC Grade 2+ to separate the subjects into low/high risk groups and drew the Kaplan Meier curves for both groups. The final survival probabilities of low and high risk groups by radiologists are 95.79% and 85.83%, respectively. For fair and direct comparison, in Fig. 3a, we selected a threshold to divide the quantified CVD risk scores using our model so that the low risk group has a survival probability of 95.79%, similar to the radiologists. Under this circumstance, the model-predicted high-risk group showed a significantly lower ($p = 0.0059$) final survival probability of 73.24%. Similarly, in Fig. 3b, we selected a threshold so that the high-risk group has a survival probability of 85.75%, also similar to the radiologists. In this case, the model-predicted low-risk group achieved a significantly higher ($p = 0.0272$) final survival probability of 97.46%. Thus, our model can help reduce inter-and intra-observer variations in quantifying CAC. The model can also automatically categorize CVD risks so that radiologists can focus on other tasks such as lung nodule detection, measurement, stability assessment, classification (based on nodule attenuation), and other incidental findings in the chest and upper abdomen. ”

- 5) In the MGH dataset, the algorithm was evaluated against various cardiovascular risk scores from the ECT-gated cardiac CTs, but not evaluated against actual presence of cardiovascular disease or related mortality. I believe that evaluation against mortality might be practically impossible, however, performance against the presence of cardiovascular disease (as in the NLST) may help readers to understand the true performance of algorithm in an independent dataset.

Response: We agree! We have made our best efforts to expand this dataset from 106 to 335 patients, including 181 CVD positive and 154 CVD negative patients. On this expanded dataset, we have conducted a new experiment of CVD screening by directly applying the model trained on the NLST dataset without any tuning. The updated results from this experiment are shown in Fig. 4a. The detailed analysis has been reported on Page 7 Paragraph 3:

“In the experiment of CVD screening shown in Fig. 4a, our deep learning model achieved a significantly higher ($p < 0.0001$) AUC value of 0.924 (95% confidence interval, 0.909-0.940) than its performance on the NLST dataset (0.871), where the network was originally trained. Superior performance on this external MGH dataset may be due to the better image quality of the contemporary MGH dataset acquired from newer scanners. Furthermore, the MGH annotated datasets and use of ECG-gated cardiac CT as a gold standard may have been superior to the annotation in NLST. This experiment shows that the proposed deep learning model has good generalizability and is highly consistent with human experts' joint label for CVD screening from LDCT and CVD scores calculated on ECG-gated cardiac CT.”

Figure 4: **Results of 4 experiments on the MGH dataset.** **a**, Validation of CVD screening. **b & c & d**, Comparison of our deep learning model with three clinical standard criteria the CAC score³³ (**b**), CAD-RADS³⁴ (**c**), and MESA 10-year risk score³⁵ (**d**) calculated with the standard protocol. For radiologists' CAC grades in **b**, red dots represents CAC Grade 1+, yellow dots represents CAC Grade 2+, green dots represents CAC Grade 3.

The description of the this expanded MGH dataset is on Page 4 Paragraph 4:

“Furthermore, through an institutional review board (IRB) approved retrospective study, we acquired an independent and fully de-identified dataset from MGH in 2020. This MGH dataset contains 335 patients (161 men, 174 women, mean age 63.6 ± 8.0 years), who underwent LDCT for lung cancer screening. Of these, 100 patients had no observed CVD abnormalities in their LDCT. The remaining 235 subjects underwent ECG-gated cardiac CT for CVD risk assessment for atypical chest pain, equivocal stress test, and chest pain with low to intermediate risk for CHD. Three CVD risk scores were calculated for the 235 subjects from their cardiac CT images, including CAC score³³, coronary stenosis (quantified as CAD-RADS)³⁴, and MESA 10-year CHD risk³⁵. Table 1 lists the characteristics of the dataset (see Methods, MGH dataset).

The MGH dataset was used to evaluate the clinical significance of the NLST-trained model for feature extraction without re-training or fine-tuning. For the validation of CVD screening, a subject was labeled as CVD-positive if the subject underwent an ECG-gated cardiac CT screening and received either a CAC score > 10 or a CAD-RADS score > 1 . Correspondingly, a CVD-negative subject had either all the LDCT screening exams being negative or the calculated scores of $CAC \leq 10$ and $CAD-RADS \leq 1$. Based on the above criteria, 181 subjects were CVD positive and 154 patients were CVD negative. For the quantification of CVD mortality risk, since this MGH dataset was collected from a patient cohort, who had recent exams (2015-2020), the mortality records are not yet available. We instead calibrate our model against the three gold standard risk scores as surrogate evaluators.”

We still preserved the original three experiments, since these three experiments evaluates the model’s reliability in CVD mortality risk estimation while the newly added experiment focuses on CVD screening. Based on the expanded dataset, the three original experiments have been updated. The corresponding details and results have been described as follows (from Page 9 to Page 10):

“To evaluate the generalization ability of the deep learning quantified CVD risk score, we directly applied the trained CVD feature extractor to the MGH data and used the extracted high-dimensional feature to estimate the three gold standard risk scores calculated from ECG-gated cardiac CT for comparison through linear classifiers (logistic regression). Five-fold cross-validation was used in all three experiments to fit and validate the three linear classifiers. Note that in all three experiments, the CVD feature extractor parameters are fixed, and the linear classifiers are directly applied to the entire high-dimensional feature without dimension reduction or feature selection. It is a standard procedure to evaluate whether the feature extractor can effectively extract features highly relative to downstream tasks³⁶⁻³⁸. Our model was compared with the radiologists’ annotated CAC grades and the two other previously reported studies on CVD risk prediction^{31,32}. Like in the NLST experiments, the CAC grades performance under the settings of 1+, 2+ and 3 was calculated. Our experimental results are presented as follows.

The features extracted by the trained CVD feature extractor from LDCT were first used to estimate the CAC score³³. With a threshold of 400 for CAC scores, the MGH subjects were divided into two groups: 78 subjects with severe CAC and 157 subjects with none or minor CAC. Our model achieved an AUC value of 0.942 (95% confidence interval, 0.927-0.958) and significantly outperformed the other two methods ($p < 0.0001$, see Fig. 4b). It is worth noting that our model is competitive with experienced radiologists. The AUC of our DL model was only slightly lower than the radiologists’ CAC grading without significant difference ($p > 0.43$), despite the fact that our DL model has never been trained for CAC score estimation. These results suggest that deep learning analysis of LDCT can well approximate the human expert performance using CCT in differentiating patients with severe and non-severe CAC.

The second experiment evaluates the capability of the deep learning model in classifying subjects into high and low risk groups using LDCT by comparing against the coronary stenosis (CAD-RADS) scores³⁴ obtained by human experts on CCT. Subjects with CAD-RADS scores greater than or equal to 4 are labeled as with severe stenosis, i.e., positive samples (51 subjects). The other 184 subjects with smaller scores were labeled as negative. Our model reached an AUC value of 0.808 (95% confidence interval, 0.758-0.858, see Fig. 4c). Our model significantly outperformed the other two methods ($p \leq 0.0338$). Unlike calcification, coronary stenosis is much harder to detect through a chest LDCT screening, while it is a direct

biomarker of CVD risk. The performance obtained using LDCT is thus highly encouraging. The superiority demonstrates that our model can quantify the subclinical imaging markers on LDCT, making it a promising tool for CVD assessment in lung cancer screening.

In the third experiment, patients were divided into high and low risk groups according to MESA 10-year risk score³⁵, which is a clinical gold-standard risk stratification score for CVD integrating multiple factors including gender, age, race, smoking habit, family history, diabetes, lipid lowering and hypertension medication, CAC score extracted from CCT, and laboratory findings including cholesterol and blood pressure. Because some of the 235 subjects did not have all the needed exams, we are only able to calculate the MESA scores of 106 subjects. When median MESA 10-year risk score in our patients was used as a threshold (14.2), 52 subjects with greater scores were labeled as high risk, while the other 54 subjects were labeled as low risk. Our model achieved an AUC value of 0.799 (95% confidence interval, 0.736-0.863), which significantly outperformed all the other methods (see Fig. 4d).”

- 6) Cardiac motion artifact and noise are major cause of limited evaluation of coronary artery calcification in the LDCTs, and their degrees are directly related with scanning and reconstruction protocols. Therefore, the authors may present detailed scanning and reconstruction protocols (e.g. CT scanner, pitch, reconstruction kernel, ...) of LDCTs (especially in the MGH dataset).

Response: Thanks for the suggestion. We have included Tables 1, Supplementary Table 2 and Supplementary Table 3 to show the scanning and reconstruction protocols for the NLST and MGH datasets, respectively.

Table 1: Characteristics of the two independent datasets used in our study.

Dataset	NLST LDCT	MGH	
		LDCT	Cardiac CT
No. of Subjects / Exams	10,395 / 21,862	335	235
Men	6,115 / 12,531	161	125
Women	4,280 / 9,331	174	110
Age (years)	61.4 ± 5.0	63.6 ± 8.0	64.9 ± 7.8
Weight (kg)	81.3 ± 18.0	82.9 ± 19.0	83.6 ± 18.5
CT Scan Setting			
Tube Voltage (kVp)	120 / 130 / 140	120 / 100	120
Milliamperere-seconds (mAs)	58.06 ± 20.59	33.80 ± 11.53	37.44 ± 10.06
Slice Thickness (mm)	2.3 ± 0.4	1.0 / 1.25	3.0
Slice Overlap (mm)	2.0 ± 0.4	0.8 / 1.0	1.5
In-plane Resolution (mm)	0.66 ± 0.07	0.83 ± 0.09	0.34 ± 0.04
Helical Pitch	1.4 ± 0.2	1.1 ± 0.2	Axial Mode

Supplementary Table 2: Manufacturers and scanner models used in the NLST dataset.

Manufacturer & Model	Volume Number	Reconstruction Kernels
GE MEDICAL SYSTEMS: CT scan	18	1. LUNG, 2 . BONE, 3 . STANDARD, 4 . BODY FILTER/STANDARD
GE MEDICAL SYSTEMS: Discovery LS	154	
GE MEDICAL SYSTEMS: Discovery QX/i	135	
GE MEDICAL SYSTEMS: HiSpeed QX/i	1596	
GE MEDICAL SYSTEMS: LightSpeed Plus	2378	
GE MEDICAL SYSTEMS: LightSpeed Power	14	
GE MEDICAL SYSTEMS: LightSpeed Pro 16	1529	
GE MEDICAL SYSTEMS: LightSpeed QX/i	4822	
GE MEDICAL SYSTEMS: LightSpeed Ultra	2491	
GE MEDICAL SYSTEMS: LightSpeed VCT	6	
GE MEDICAL SYSTEMS: LightSpeed16	3872	
GE MEDICAL SYSTEMS: QX/i	3	
Philips: Mx8000	2382	1. D, 2. C, 3. B
Philips: Mx8000 IDT	90	
Philips: Mx8000 IDT 16	65	
SIEMENS: Emotion 16	14	1. B50f, 2. B45f, 3. B50s, 4. B60f, 5. B60s, 6. B70f, 7. B30f, 8. B31s, 9. B80f, 10. B20f, 11. B30s, 12. B31f
SIEMENS: Emotion 6	3	
SIEMENS: Sensation 10	2	
SIEMENS: Sensation 16	3870	
SIEMENS: Sensation 4	706	
SIEMENS: Sensation 64	393	
SIEMENS: Volume Zoom	7001	
TOSHIBA: Aquilion	1869	1. FC51, 2. FC50, 3. FC53, 4. FC30, 5. FC10, 6. FC82, 7. FC02, 8. FC01
Summary	33413	

Supplementary Table 3: Manufacturers and scanner models used in the MGH dataset.

Manufacturer & Model	Volume Number	Reconstruction Kernels
LDCT		
GE MEDICAL SYSTEMS: Discovery CT750 HD	68	1. STANDARD
GE MEDICAL SYSTEMS: Discovery STE	13	
GE MEDICAL SYSTEMS: LightSpeed Pro 16	18	
GE MEDICAL SYSTEMS: LightSpeed VCT	29	
GE MEDICAL SYSTEMS: Revolution CT	51	
Philips : Brilliance 40	5	1. B
Philips : iCT 256	42	
Philips : IQon - Spectral CT	57	
SIEMENS : Biograph 64	3	1. I30f\3, 2. I30f\2, 3. Br40d\3, 4. I31f\3, 5. B30f, 6. Br40f\2
SIEMENS : SOMATOM Definition Edge	35	
SIEMENS : SOMATOM Definition Flash	4	
SIEMENS : SOMATOM Drive	2	
SIEMENS : SOMATOM Force	8	
Summary	335	
ECG-gated CCT		
SIEMENS : SOMATOM Force	84	1. B35f, 2. B31f
SIEMENS : SOMATOM Definition Flash	151	
Summary	235	

7) *Although I am not a statistical expert, I am concerned about whether the ROC analyses for cardiovascular mortality ignoring the time to death and censoring is appropriate.*

Response: Thanks for the comment. To appropriately evaluate our model for CVD mortality prediction, we have included the Kaplan Meier curves of the CVD-caused death in Fig. 3. The corresponding details have been provided in the response to comment #4) above.

Reviewers' Comments:

Reviewer #1:

Remarks to the Author:

The authors have made significant revisions. The Kaplan Meier plots are helpful. They have incorporated the sensitivity and PPV. The distinction in the endpoints is helpful. They comment on deaths from unknown causes. It is nice that the responses, edits, and figures are in the response letter.

I just have a few minor questions, maybe the authors could clarify:

1. I understand there are 2 endpoints, diagnosed CVDs and CVD related mortality. If patients are being followed by LDCTs, then by definition, they have risk factors for lung cancer (usually significant smoking history or tobacco use), and these same risk factors also cause CVDs. Thus, they are likely followed by lung and heart physicians, and by default, they will be more likely to be labeled as having a CVD by either provider. Can the authors better describe this in the paper? Would it affect their findings?
2. In the KM plots, what is "reader"? Is that the same thing as radiologist? Can you provide the number at risk below? Can you make the x axis use good time points (eg 12, 24 months; or years). I am not sure what time is and why it is going by 500s. I would make the y axis go from 0-1. Can you provide a p value comparison among the curves? What is the survival probability (y axis)? Death from any cause? In Figure 2, "Reader" is also used, and you may want to use radiologist or reader to be consistent.
3. Thank you for better explaining the model (https://colab.research.google.com/github/DIAL-RPI/CVD-Risk-Estimator/blob/develop/colab_run.ipynb) and providing Figure 1 and Sup Figure 1. For clinicians, this will still be a black box. For example, clinicians do not understand CNN, Tri2D-Net, extracted high dimensional CVD feature, deep CNN (is that different from CNN?), gradient weight class activation mapping. Other terms that clinicians may not know are MESA, CAC, CAD-RADS, AE+SVM, KAMP NET, Grad-CAM. You may want to explain it a little better for people with no understanding of machine learning or these minutiae of this field. Consider using a brief glossary or table.
4. The authors could incorporate a table of patient demographics (similar to 1, but centered on the patients and the scans). How many exams were "normal" vs "abnormal"? What are some of the features detected in them? What was the comorbidity breakdown?

Reviewer #2:

Remarks to the Author:

This revision of the original paper contains several important improvements over the original paper. The most important improvements are the extension of the MGH validation set and the extension of the experiments, especially the Kaplan-Meier curves. In the meantime, a paper with a similar task has been published in Nature Communications:

<https://www.nature.com/articles/s41467-021-20966-2>. I think the paper has improved substantially, but I still have several major comments:

1. I think it would make a lot of sense if this paper is put into context with the already published paper. That code is also publicly available, so it would be a much better comparison than the current comparison with the AE-SVM and KAMP-Net in my opinion.
2. The paper is still confusing to me because it tackles both CVD screening (findings CVD abnormalities, primarily CAC), and CVD mortality prediction. For mortality prediction, there is no external data set, because there is no mortality data present for the MGH dataset. Therefore, the paper has little evidence on how the presented system would generalize to other datasets for predicting mortality.
2. To be able to use the presented system for CVD screening on the MGH dataset, the authors

have to perform a 5-fold CV where they use linear classifiers on the features extracted from the original network as the final classifiers. It is unclear what linear classifiers are used. This needs clarification. Are they applied on the output of the fully connected layer, so the 1536-dimensional feature vector? If the authors used 5-fold CV, and 4 folds are used to optimize the linear classifier and 1 fold as test, this means that there are roughly 270 cases used to optimize this linear classifier? How different were the 5 classifiers in the end? This gives an indication how stable the feature set is. For me, it would be much more clear if the score of the original system would be validated in these experiments. This would then show the readers how the output of the system trained on NLST corresponds to CAC scores on the MGH dataset.

- One of the claims of the paper is that this system is trained end-to-end for mortality prediction and hence may look at more imaging aspects than only CAC. For example, it could look at epicardial fat. There is very little evidence that this is really the case. The only evidence I can find is the anecdotal Grad-CAM images from Figure 5. An experiment in which radiologists label this, and then a check on how the scores of the system correspond with that, would be a step to do that.

Minor comments:

- I now understand that a random CT scan per patient is used for the NLST data. Also, I think I now understand that patients with inconsistent CVD labels over time are excluded. Therefore, the authors go from 16,264 subjects to 10,395 subjects. Two comments: 1) I think it would have made more sense if the baseline CT would be used for all NLST subjects, because then, you had roughly 6.5 years of follow-up for all NLST subjects. But this is a minor comment. 2) The exclusion of subjects with inconsistent labels is a limitation of the study that needs to be added to the discussion.

Reviewer #3:

Remarks to the Author:

I appreciate to have a chance to review the revised manuscript.

I believe the concerns brought up in the initial manuscript were substantially reduced.

Response to Reviewer#1:

The authors have made significant revisions. The Kaplan Meier plots are helpful. They have incorporated the sensitivity and PPV. The distinction in the endpoints is helpful. They comment on deaths from unknown causes. it is nice that the responses, edits, and figures are in the response letter.

Response: We would like to thank the reviewer for the great comments that have guided us in significantly improving the quality of the paper.

I just have a few minor questions, maybe the authors could clarify:

1.1) *I understand there are 2 endpoints, diagnosed CVDs and CVD related mortality. If patients are being followed by LDCTs, then by definition, they have risk factors for lung cancer (usually significant smoking history or tobacco use), and these same risk factors also cause CVDs. Thus, they are likely followed by lung and heart physicians, and by default, they will be more likely to be labeled as having a CVD by either provider. Can the authors better describe this in the paper? Would it affect their findings?*

Response: In a recent study of 3,110 patients who underwent LDCT for lung cancer screening, contrary to the general perception, less than one-third of patients with significant CAC (236/756 patients) had established diagnosis of coronary artery disease at the baseline LDCT [7]. Furthermore, of the 155 patients with a change in cardiovascular management, under one-quarter of patients (36/155 patients) were referred to cardiologists. Given the fact that patients undergoing LDCT are asymptomatic subjects who can undergo definitive lung cancer treatment, it is desirable and beneficial that an effective CVD risk estimating program helps uncover those patients who are at higher risk of CVD-related mortality. This demonstrates the potential for our algorithm to better assess the risk of CVD and trigger the next level of mitigating steps to decrease the associated risk. We have revised the following paragraph on page 1 in the paper to clarify it.

“Cardiovascular disease (CVD) affects nearly half of American adults and causes more than 30% of fatality¹. The prediction of CVD risk is fundamental to the clinical practice in managing patient health². Recent studies have shown that the patients diagnosed with cancer have a ten-fold greater risk of CVD mortality than the general population³. For lung cancer screening, low dose computed tomography (LDCT) has been proven effective through clinical trials^{4, 5}. In the National Lung Screening Trial (NLST), 356 participants who underwent LDCT died of lung cancer during the 6-year follow-up period. However, more patients, 486 others, died of CVD. The NELSON trial shows similar overall mortality rates between the study groups even though the lung cancer mortality decreased in the LDCT screening group⁶. Therefore, screening significant comorbidities like CVD in high-risk subjects undergoing LDCT for lung cancer screening is critical to lower the overall mortality. Nevertheless, when the cancer risk population receives cancer screening, their potential CVD risk may be overlooked. A recent study reported that only one-third of patients with significant coronary artery calcification (CAC) on LDCT had established coronary artery disease diagnosis, whereas just under one-quarter of the patients had a change in his/her cardiovascular management following LDCT with referral to cardiologists⁷.”

1.2) *In the KM plots, what is “reader”? Is that the same thing as radiologist? Can you provide the number at risk below? Can you make the x axis use good time points (eg*

12, 24 months; or years). I am not sure what time is and why it is going by 500s. I would make the y axis go from 0-1. Can you provide a p value comparison among the curves? What is the survival probability (y axis)? Death from any cause? In Figure 2, “Reader” is also used, and you may want to use radiologist or reader to be consistent.

Response: By “reader” we meant “radiologist”. We apologize for the confusion. In our revised paper, we have replaced “reader” with “radiologist” in Fig. 3.

We have updated the KM plots in Fig. 3 by clearly stating the numbers at risk. We have changed the x-axis label from “number of days” to “number of years” for clarity. The y-axis has been normalized. Since all the datapoints are above 0.7, to better visualize the data and show the differences, the y-axis tick starts from 0.7. It has been made consistent across the two KM plots. We have also clarified the label for the y axis to be “CVD-death free survival”, and added p-values between the curves in Fig. 3, as shown below.

Figure 3: **Kaplan Meier curves on the NLST dataset.** Comparison of our model with the radiologists. Thresholds were selected on the CVD risk score calculated by our model to enforce the low (a) or high (b) risk group has a similar survival probability with that of the radiologists.

1.3) Thank you for better explaining the model (https://colab.research.google.com/github/DIAL-RPI/CVD-Risk-Estimator/blob/develop/colab_run.ipynb) and providing Figure 1 and Sup Figure 1. For clinicians, this will still be a black box. For example, clinicians do not understand CNN, Tri2D-Net, extracted high dimensional CVD feature, deep CNN (is that different from CNN?), gradient weight class activation mapping. Other terms that clinicians may not know are MESA, CAC, CAD-RADS, AE+SVM, KAMP NET, Grad-CAM. You may want to explain it a little better for people with no understanding of machine learning or these minutiae of this field. Consider using a brief glossary or table.

Response: Great point! We have provided a glossary in the supplementary material on pages S1 and S2, which is also attached below.

“CAC Score: The coronary artery calcium (CAC) score is a semiquantitative measure of coronary calcification with ECG-gated, non-contrast CT. Agatston score³⁴ is used as a measure of CAC in this paper. It reflects the total area of calcium deposits and the density of the calcium in coronary artery.

CAD-RADS: The Coronary Artery Disease - Reporting and Data System (CAD-RADS)³⁵ is an expert consensus document developed to standardize reporting of findings with coronary CT angiography.

MESA Score: The Multi-Ethnic Study of Atherosclerosis (MESA) risk score³⁶ is an estimation of 10-year coronary heart disease risk obtained using traditional risk factors and coronary artery calcium.

CNN (a.k.a. deep CNN): A convolutional neural network (CNN) is one class of deep neural networks that most commonly applied to visual images analysis.

DeepCAC: A deep learning based system¹⁷ designed for automatically calculating the CAC score from a chest CT image.

AE+SVM: A two-stage machine learning based model³² designed for CVD mortality prediction. It is composed by an auto-encoder (AE) for image feature extraction and a support vector machine (SVM) for classification.

KAMP-Net: An end-to-end deep learning based model³³ designed for all-cause mortality prediction from a low-dose chest CT scan.

Grad-CAM: Gradient-weighted Class Activation Mapping (Grad-CAM)³⁷ is a visualization approach for intuitive interpretation of decisions made by a convolutional neural network based model. It uses the gradient of a target class flowing into the final convolutional layer to produce a coarse localization map highlighting important regions in an image for predicting the class.”

1.4) The authors could incorporate a table of patient demographics (similar to 1, but centered on the patients and the scans). How many exams were “normal” vs “abnormal”? What are some of the features detected in them? What was the comorbidity breakdown?

Response: To present patient demographics more clearly, we have split the original Table 1 into new Tables 1 and 2 focusing on demographics and scans parameters, respectively. We have added the number of “normal” and “abnormal” LDCT exams to Table 1. The new tables are on pages 4 and 5.

In addition, we have provided additional information on comorbidity breakdown in Supplementary Figs. 4-6.

All the new tables and figures are also attached below for your convenience.

Table 1: Demographic of the two independent datasets used in our study.

Dataset	NLST LDCT			MGH	
	CVD Screening Positive/Negative	CVD-Related Deaths/Survival	Overall	LDCT	Cardiac CT
# Patients	2,962 / 7433	600 / 9795	10,395	335	235
Men	2,048 / 4,067	442 / 5,673	6,115	161	125
Women	914 / 3,366	158 / 4,122	4,280	174	110
Age (years)	62.9 ± 5.3 / 60.8 ± 4.8	64.0 ± 5.5 / 61.3 ± 5.0	61.4 ± 5.0	63.6 ± 8.0	64.9 ± 7.8
Weight (kg)	85.9 ± 19.3 / 79.4 ± 17.1	86.5 ± 21 / 81.0 ± 17.7	81.3 ± 18.0	82.9 ± 19.0	83.6 ± 18.5

Table 2: CT Scan characteristics of the two independent datasets used in our study.

Dataset	NLST LDCT	MGH	
		LDCT	Cardiac CT
Tube Voltage (kVp)	120 / 130 / 140	120 / 100	120
Milliamperere-seconds (mAs)	58.06 ± 20.59	33.80 ± 11.53	37.44 ± 10.06
Slice Thickness (mm)	2.3 ± 0.4	1.0 / 1.25	3.0
Slice Overlap (mm)	2.0 ± 0.4	0.8 / 1.0	1.5
In-plane Resolution (mm)	0.66 ± 0.07	0.83 ± 0.09	0.34 ± 0.04
Helical Pitch	1.4 ± 0.2	1.1 ± 0.2	Axial Mode

Supplementary Fig. 4: **Causes of death co-occurrence matrix.** This matrix describes the co-occurrence of 5 major CVD mortality causes (rows) and 17 overall major causes of death (columns) in NLST. For instance, the number 58 in the first row of the first column indicates that 58 deceased patients had both *essential (primary) hypertension (I10)* and *chronic ischemic heart disease (I25)* listed as their causes of death.

Supplementary Fig. 5: **Histogram of major causes of death breakdown in the NLST dataset.** Note that a patient might die from multiple causes.

Supplementary Fig. 6: **Histogram of the 17 CVD related causes of death in the NLST dataset.**
 Note that a patient might die from multiple causes.

Response to Reviewer#2:

This revision of the original paper contains several important improvements over the original paper. The most important improvements are the extension of the MGH validation set and the extension of the experiments, especially the Kaplan-Meier curves. In the meantime, a paper with a similar task has been published in Nature Communications: <https://www.nature.com/articles/s41467-021-20966-2>.

Response: We would like to thank you so much for helping us improve our work and also pointing out the recently published paper in Nature Communications on a very similar topic. It is a great opportunity for us to use the results in that paper for not only cross validating our work but also demonstrating the unique values of our approach and results. We have therefore included the results obtained using their shared code in this revised version. More details are provided in the point-to-point response below.

● Major Comments

I think the paper has improved substantially, but I still have several major comments:

2.1) *I think it would make a lot of sense if this paper is put into context with the already published paper. That code is also publicly available, so it would be a much better comparison than the current comparison with the AE-SVM and KAMP-Net in my opinion.*

Response: We completely agree with you and have thus included the comparison with the work (<https://www.nature.com/articles/s41467-021-20966-2>), which is referred to as DeepCAC [17] in the revised version. DeepCAC developed a series of deep convolutional neural networks to locate the heart and then segment coronary artery calcium (CAC) from CT images. The prediction of cardiovascular risk is solely based on the quantified CAC. In contrast, after the heart is separated using a detection network, our deep network directly predicts the cardiovascular risk using the automatically learnt features including but not limited to CAC.

We have included the ROCs obtained using the source code released by the authors of [17] in Figs. 4 and 2 in the revised version. As is clearly shown, DeepCAC indeed performed better than the two previous methods, AE+SVM and KAMP-Net. However, our proposed method outperformed DeepCAC with a significant margin ($p \leq 0.0017$) in both cardiovascular disease (CVD) screening and CVD mortality prediction on the NLST dataset. This is because our proposed method exploits other imaging features in addition to CAC, such as paracardiac fat.

We have also applied DeepCAC to the MGH dataset for CVD screening. Again, our model significantly outperformed DeepCAC ($p=0.0102$). Note that DeepCAC also performed better on the MGH dataset (AUC=0.879) than the NLST dataset (AUC=0.753), which is consistent with our model (AUC=0.924 on the MGH dataset and AUC=0.871 on the NLST dataset). As we have explained in the manuscript, this may be due to the fact that the MGH dataset has better CT image quality.

All the revised figures and corresponding analysis are also attached below for your convenience.

Figure 2: **Experimental results on the NLST dataset.** **a**, Reader study and comparison of our model with other reported methods on CVD detection. **b**, Reader study and comparison of our model with other reported methods on CVD caused mortality prediction. For reader study, green symbols represents CAC Grade 1+, yellow symbols represents CAC Grade 2+, red symbols represents CAC Grade 3.

“We first evaluated the proposed model for identifying patients with CVDs from the lung cancer screening population. Fig. 2a shows the receiver operating characteristic curves (ROCs) of multiple methods. Our deep learning model achieved an area under the curve (AUC) of 0.871 (95% confidence interval, 0.860-0.882)(see Methods, Statistical analysis). With a positive predictive value (PPV) of 50.00%, the model achieved a sensitivity of 87.69%, which suggests that our model can identify 87.69% of the CVD-positive subjects using only a chest LDCT scan, when allowing half of the positive predictions as false. For the radiologist performance, all patients with \geq minimal CAC (CAC Grade 1+) are considered as abnormal. It can be seen in Fig. 2a that CAC Grade 1+ yielded a sensitivity of 96.6% and a PPV of 35.3%. With a similar sensitivity of 96.6%, our model achieved a slightly but not significantly higher PPV of 38.4% ($p=0.3847$). In addition, we compared our model with the three recently reported works, KAMP-Net³³, Auto-encoder (AE+SVM)³², and a deep learning based CAC scoring model (DeepCAC)¹⁷. The table in Fig. 2a shows that our model significantly outperformed the other three methods ($p<0.0001$). It indicates that our model can use LDCT to differentiate subjects with high CVD risk from those with low risk.

Further, we evaluated the performance of our model in quantifying CVD mortality risk. The results are shown in Fig. 2b, where CAC Grades 1+, 2+, and 3 denote the performance of mortality prediction using extent of subjective CAC categories 1 and above, categories 2 and above, and 3 only, respectively. The trained deep learning model was directly applied to this testing set to predict the CVD-caused mortality without fine-tuning. Our deep learning model achieved an AUC value of 0.768 (95% confidence interval, 0.734-0.801), which significantly outperformed the competing methods ($p \leq 0.0017$) as shown in Fig. 2b. With the same PPV of averaged CAC Grade 2+ (10.8%), our model achieved a sensitivity of 80.8%. Specifically, in the NLST test set, our model successfully identified 97 of the 120 deceased subjects as high

risk, while the averaged CAC Grade 2+ labeled 35 of those 97 cases as low risk. Additionally, it can be seen from Fig. 2b that our model achieved a similar performance to the average performance of human experts. It is worth mentioning that a significant difference exists between the three radiologists' annotations ($p < 0.0001$), even though radiologist 1 performed better than our model.”

Figure 4: **Results of the four experiments on the MGH dataset.** **a**, Validation of CVD screening. **b & c & d**, Comparison of our deep learning model with three clinical standard criteria the CAC score³⁴ (**b**), CAD-RADS³⁵ (**c**), and MESA 10-year risk score³⁶ (**d**) calculated with the standard protocol. For radiologists' CAC grades in **b**, red dots represents CAC Grade 1+, yellow dots represents CAC Grade 2+, green dots represents CAC Grade 3.

“In the experiment of CVD screening shown in Fig. 4a, our deep learning model achieved a significantly higher ($p < 0.0001$) AUC value of 0.924 (95% confidence interval, 0.909-0.940) than its performance on the NLST dataset (0.871), where the network was originally trained. Superior performance on this external MGH dataset may be due to the following two factors. First, MGH dataset acquired with contemporary scanners contains better quality images.

Second, the MGH dataset combines the annotations of LDCT and ECG-gated cardiac CT as gold standard, which is more accurate than the annotation in NLST.”

- 2.2) *The paper is still confusing to me because it tackles both CVD screening (findings CVD abnormalities, primarily CAC), and CVD mortality prediction. For mortality prediction, there is no external data set, because there is no mortality data present for the MGH dataset. Therefore, the paper has little evidence on how the presented system would generalize to other datasets for predicting mortality.*

Response: The performance of CVD mortality prediction shown in Fig. 2b was evaluated on an independent test set from NLST, which was separated from the training and validation sets. The test data subjects were included in neither the training nor validation sets. In the recent DeepCAC paper [17], the method was also validated on the NLST dataset for mortality prediction.

We agree with the reviewer that the lack of mortality recorded imaging datasets in both public and private domains is a general limitation of the existing studies. Since the use of LDCT at MGH (as well as from other US sites) started after the USPTF recommendations on LDCT, which came much later than the NLST, we do not have access to a sufficient number of patients with mortality information from CVD. We believe that with the increasing use of LDCT over time, CT data with CVD mortality information will become available and allow us to test generalizability of our work on datasets beyond NLST. The corresponding discussion has been included in Paragraph 2 on Page 11:

“Our study’s limitation includes using CVD related ICD-10 codes for labeling the subjects, which may miss some CVD-related deaths or mislabel patients who died from other heart diseases as CVD mortality. Mitigating its influence on our study motivated us to collect data at MGH and evaluate our model using the surrogate gold standard CVD risk scores on the MGH dataset. Our results on the MGH dataset support the pre-trained model’s utilities on data from a different source. Another limitation is that we did not have access to a sufficient number of patients with mortality information from CVD. The use of LDCT at MGH (as well as from other United States sites) started after the United States Preventive Services Taskforce recommended annual screening for lung cancer with LDCT in 2013. We believe that with the increasing use of LDCT over time, CT data with CVD mortality information will become available.”

- 2.3) *To be able to use the presented system for CVD screening on the MGH dataset, the authors have to perform a 5-fold CV where they use linear classifiers on the features extracted from the original network as the final classifiers. It is unclear what linear classifiers are used. This needs clarification. Are they applied on the output of the fully connected layer, so the 1536-dimensional feature vector? If the authors used 5-fold CV, and 4 folds are used to optimize the linear classifier and 1 fold as test, this means that there are roughly 270 cases used to optimize this linear classifier? How different were the 5 classifiers in the end? This gives an indication how stable the feature set is. For me, it would be much more clear if the score of the original system would be validated in these experiments. This would then show the readers how the output of the system trained on NLST corresponds to CAC scores on the MGH dataset.*

Response: We agree with the reviewer that directly validating the scores of our model in those experiments on the MGH dataset would be clearer. This was the case for the CVD screening, but not for the other three experiments. In this revision, we have thus followed the reviewer’s suggestion and re-performed the experiments to directly apply our model output to predict the

risks categorized by CAC score, CAD-RADS, and MESA 10-years risk score, respectively. We have also updated Fig. 4 to include the new experimental results. The figure is attached under the response to Comment #2.1. We have also revised the corresponding description in the main text (Pages 9 and 10), which is reproduced as follows:

“To evaluate the generalization ability of the deep learning quantified CVD risk score, we directly applied the trained model to the MGH data and evaluate the consistency between the model predicted risk score from LDCT and the three clinically adopted risk scores calculated from ECG-gated cardiac CT. Our model was compared with two other previously reported studies on CVD risk prediction^{32,33}.

The predicted risk score from LDCT was first evaluated against the CAC score³⁴. With a threshold of 400 for CAC scores, the MGH subjects were divided into two groups: 78 subjects with severe CAC and 157 subjects with none or minor CAC. Our model achieved an AUC value of 0.881 (95% confidence interval, 0.851-0.910) and significantly outperformed the other two methods ($p < 0.0001$, see Fig. 4b), despite the fact that our model has never been trained for CAC score estimation. These results suggest that our deep learning quantified CVD risk score is highly consistent with the CAC scores derived from ECG-gated cardiac CT in differentiating patients with severe and non-severe CAC.

The second experiment evaluates the capability of the deep learning model in classifying subjects into high and low risk groups using LDCT by comparing against the coronary stenosis (CAD-RADS) scores³⁵ obtained by human experts on CCT. Subjects with CAD-RADS scores greater than or equal to 4 are labeled as with severe stenosis, i.e., positive samples (51 subjects). The other 184 subjects with smaller scores were labeled as negative. Our model reached an AUC value of 0.763 (95% confidence interval, 0.704-0.821, see Fig. 4c). Our model significantly outperformed the other two methods ($p \leq 0.0080$). Unlike calcification, coronary stenosis is much harder to detect through a chest LDCT screening, while it is a direct biomarker of CVD risk. The performance obtained using LDCT is thus highly encouraging. The superiority demonstrates that our model can quantify the subclinical imaging markers on LDCT, making it a promising tool for CVD assessment in lung cancer screening.

In the third experiment, patients were divided into high and low risk groups according to MESA 10-year risk score³⁶, which is a clinical gold-standard risk stratification score for CVD integrating multiple factors including gender, age, race, smoking habit, family history, diabetes, lipid lowering and hypertension medication, CAC score extracted from CCT, and laboratory findings including cholesterol and blood pressure. Because some of the 235 subjects did not have all the needed exams, we are only able to calculate the MESA scores of 106 subjects. When median MESA 10-year risk score in our patients was used as a threshold (14.2), 52 subjects with greater scores were labeled as high risk, while the other 54 subjects were labeled as low risk. Our model achieved an AUC value of 0.835 (95% confidence interval, 0.781-0.890), which significantly outperformed all the other methods (see Fig. 4d).”

2.4) *One of the claims of the paper is that this system is trained end-to-end for mortality prediction and hence may look at more imaging aspects than only CAC. For example, it could look at epicardial fat. There is very little evidence that this is really the case. The only evidence I can find is the anecdotal Grad-CAM images from Figure 5. An experiment in which radiologists label this, and then a check on how the scores of the system correspond with that, would be a step to do that.*

Response: Good point! We agree that Fig. 5 only shows three cases on epicardial fat using Grad-CAM. To demonstrate that the developed system utilizes other information besides CAC, we computed the correlation between our deep learning estimated CVD risk score and the

epicardial fat volume. To measure the epicardial fat volume, we first used an existing deep learning based heart segmentation model [17] to segment the heart region. Then, we identified fat using HU-value-based thresholding. All voxels within [-190,-30] HU were considered as fat. The Pearson correlation coefficient between our model predicted risk score and the fat volume is 0.199 ($p < 0.0001$). In contrast, the radiologist estimated CAC scores had a Pearson correlation of 0.078 ($p = 0.0004$) with the fat volume. The deep learning auto-estimated CAC by DeepCAC got a Pearson correlation of 0.0851 ($p = 0.0001$) with the fat volume. The correlation between the output of our model and the volume of fat is much stronger than the compared methods that focus on CAC only. This indicates that our model considers fat as one of risk factors in the risk quantification. The corresponding results have been included in the Discussions section of the revised manuscript.

In addition, we came across several references, which have consistently demonstrated that epicardial/pericardial fat is correlated with CVD risks. Prior studies including one by Rosito et al. on 1,155 patients from the Framingham Heart Study reported that pericardial fat was related to hypertension, diabetes mellitus, and higher triglycerides, lower high density lipoprotein, and metabolic syndromes ($p < 0.01$) [39]. In another study on 1,030 patients with type 2 diabetes, Christensen et al. reported that epicardial fat improved risk prediction over other CVD risk factors [38].

“To interpret the prediction results of Tri2D-Net, we generated heatmaps using the Gradient-weighted Class Activation Mapping (Grad-CAM)³⁷ and exported the attention maps from the attention block. Fig. 5 shows the results of three representative subjects from the NLST dataset. Figs. 5a and 5b belong to two subjects who died of CVD, referred as Case-a and Case-b, respectively. Fig. 5c shows the image of a subject Case-c, who survived by the end of the trial. Case-a has severe CAC with an average CAC Grade of 3.0 by the three radiologists. Tri2D-Net captured the strong calcium as shown in the Grad-CAM heatmap and predicted a high CVD risk score of 0.90. Case-b (Fig. 5b) had mild to moderate CAC with an average CAC Grade of 1.67. However, the attention block noticed abundant juxtacardiac fat and Tri2D-Net gave a high score of 0.82 for the case. Case-c has mild CAC graded as 1.33 by the radiologists. Since there was little calcification and juxtacardiac fat as indicated by the heatmaps, Tri2D-Net predicted a low risk score of 0.23 for this survived patient. Visualization of these cases demonstrates the contributions of both CAC and juxtacardiac fat to our model for predicting CVD risk scores in contrast with the mere reliance on CAC as the sole biomarker in prior studies^{17, 28-31}. It is consistent with the clinical findings that epicardial/pericardial fat correlates with several CVD risks^{38,39}. Based on this finding, we further examined the Pearson correlation between our model predicted CVD risk score and juxtacardiac fat volume. Our model achieved a Pearson correlation of 0.199 ($p < 0.0001$) with juxtacardiac fat volume. In contrast, the radiologist estimated CAC and the deep learning estimated CAC¹⁷ got Pearson correlation of 0.078 ($p = 0.0004$) and 0.085 ($p = 0.0001$), respectively. The ability to capture various features in addition to CAC makes our model superior to the existing CAC scoring models for CVD screening and CVD mortality quantification.”

● Minor Comments

- 1) I now understand that a random CT scan per patient is used for the NLST data. Also, I think I now understood that patients with inconsistent CVD labels over time are excluded. Therefore, the authors go from 16,264 subjects to 10,395 subjects. Two comments: 1) I think it would have made more sense if the baseline CT would be used for all NLST subjects, because then, you had roughly 6.5 years of follow-up for all NLST subjects. But this is a

minor comment. 2) The exclusion of subjects with inconsistent labels is a limitation of the study that needs to be added to the discussion.

Response: 1) When training the model, we used LDCT scans of each patient from all the available time points. In our sampled test set, 1,851 subjects were active at the end of the trial with an averaged follow-up time of 6.5 ± 0.6 years. In these subjects, only 275 subjects had follow-up time less than 6 years. The remaining 1,576 subjects had greater than 6 years of follow-up. We plotted the histogram of the follow-up time below to show the distribution.

2) We excluded 5,869 subjects due to either technical reasons (slice spacing $> 3\text{mm}$, scan length along superior to inferior $< 200\text{ mm}$, or inability to process certain LDCT images) or for the lack of information related to absence or presence of CVD. For those subjects with inconsistent labels in their LDCT reports, we did not exclude any of them but only used their scans with clear/latest labels. We have clarified this point in the revised version.

“The NLST data are publicly available through the Cancer Data Access System (CDAS) of the National Institutes of Health (NIH). LDCTs were collected from multiple institutions, with slice spacing varying from 0.5mm to 5mm. Scans with slice spacing larger than 3mm or with scan length along superior to inferior less than 200 mm were filtered out. Supplementary Fig. 2 shows the inclusion and exclusion criteria. Since the NLST was designed for lung cancer screening, CVD related information is incomplete. Therefore, we only used LDCT images with clear CVD information. Specifically, for all CVD-negative patients and CVD-positive patients who died in the trial with CVD-related causes, all available LDCT exams were considered as valid exams. For other CVD-positive patients, only the exams with clear CVD abnormalities reported were considered as valid exams. In the training set, all LDCT volumes generated in the valid exams were treated as independent cases to increase the number of training samples. In the validation and testing phases, a single CT volume was randomly selected from the valid exams of each subject to preserve the original data distribution. Since the number of LDCT exams is inconsistent across subjects (for example, patients with death or diagnosis of lung cancer on initial LDCTs did not complete all the follow-up LDCTs), the use of these CT volumes as independent cases can change the data distribution and introduce a bias to the results. After the random selection, the formed test set keeps an averaged follow-up time of

6.5±0.6 years. Other properties of the NLST dataset are summarized in Tables 1 and 2. More detailed information including manufacturer, scanner, and reconstruction kernel can be found in Supplementary Table 2.”

Response to Reviewer#3:

I appreciate to have a chance to review the revised manuscript. I believe the concerns brought up in the initial manuscript were substantially reduced.

Response: We appreciate the reviewer's critiques/suggestions very much which are invaluable in helping us improve the quality of our paper.

Reviewers' Comments:

Reviewer #1:

None

Reviewer #2:

Remarks to the Author:

I appreciate the extensive revision of the manuscript by the authors, and I think the changes have improved the manuscript a lot! The improved performance over the recently published DeepCAC model is promising.

I am satisfied with the responses, and only have two minor points:

1) I do not understand why the results in Fig 4 for KAMP-Net and AE+SVM have also changed for the CAC, CAD-RADS and MESA figures. Did the authors previously also use linear classifiers on top of the features extracted by these approaches in the 5-fold CV? That would be my only explanation..

2) I appreciate the correlation experiment with the juxtacardiac fat and I think this adds the needed evidence that the model also looks at juxtacardiac fat. From the letter to the reviewers, it is clear that an automatic algorithm is used to measure this, but it is not explained in the added section in the discussion. I think this needs to be briefly explained to the readers, just like the authors explained it to me in the letter.

Responses to Reviewers' Critiques

NCOMMS-20-33634B

Deep Learning Predicts Cardiovascular Disease Risks from Lung Cancer Screening Low-dose Computed Tomography

Hanqing Chao, Hongming Shan, Fatemeh Homayounieh, Ramandeep Singh,
Ruhani Doda Khera, Hengtao Guo, Timothy Su,
Ge Wang* , Mannudeep K. Kalra*, Pingkun Yan*

*Co-corresponding author e-mails:

wangg6@rpi.edu

mkalra@mgh.harvard.edu

yanp2@rpi.edu

Comments and Responses

AE Comment: *“Thank you again for submitting your manuscript “Deep Learning Predicts Cardiovascular Disease Risks from Lung Cancer Screening Low Dose Computed Tomography” to Nature Communications. We have now received reports from 2 reviewers and, on the basis of their comments, we have decided to invite a revision of your work for further consideration in our journal. Your revision should address all the points raised by our reviewers (see their reports below).”*

Response: We are highly grateful to the editor and the reviewers for the very quick turnaround. In this revision, we have addressed the two minor comments, as detailed in our point-to-point responses below. The manuscript has also been updated accordingly with the changes highlighted in blue.

Response to Reviewer #2:

I appreciate the extensive revision of the manuscript by the authors, and I think the changes have improved the manuscript a lot! The improved performance over the recently published DeepCAC model is promising.

Response: We appreciate the reviewer's advice, which has helped us tremendously improve the quality of our paper.

I am satisfied with the responses, and only have two minor points:

1) I do not understand why the results in Fig 4 for KAMP-Net and AE+SVM have also changed for the CAC, CAD-RADS and MESA figures. Did the authors previously also use linear classifiers on top of the features extracted by these approaches in the 5-fold CV? That would be my only explanation.

Response: Your understanding is correct. We previously used linear classifiers on top of the features extracted by these approaches in the 5-fold CV, but in our last round of revision all the algorithms were directly applied to the MGH datasets without finetuning. Hence, the results were accordingly changed in the updated figures.

2) I appreciate the correlation experiment with the juxtacardiac fat and I think this adds the needed evidence that the model also looks at juxtacardiac fat. From the letter to the reviewers, it is clear that an automatic algorithm is used to measure this, but it is not explained in the added section in the discussion. I think this needs to be briefly explained to the readers, just like the authors explained it to me in the letter.

Response: Good point! We have added the explanation in the first paragraph on page 11 of the main text.

“To measure the volume of juxtacardiac fat, we first used an existing deep learning based heart segmentation model¹⁷ to segment the heart region. Then, we identified fat inside the segmented heart by HU-based cutoffs (voxels in [-190,-30] HU were considered to represent fat).”

Reviewers' Comments:

Reviewer #2:

Remarks to the Author:

I am satisfied with the responses of the authors. I think the paper is ready for publication.

Response to Reviewer #2:

I am satisfied with the responses of the authors. I think the paper is ready for publication.

Response: We highly appreciate your effort and advice that has helped us significantly enhance the quality and impact of our work.